# RNA splicing analysis using heterogeneous and large RNA-seq datasets

Jorge Vaquero-Garcia[1,5], Joseph K. Aicher [1,2,5], San Jewell[1,5], Matthew R. Gazzara [1,5], Caleb M. Radens[1], Anupama Jha[3], Scott S. Norton [1], Nicholas F. Lahens [4], Gregory R. Grant[1,4] & Yoseph Barash [1,3] ✉

The ubiquity of RNA-seq has led to many methods that use RNA-seq data to analyze variations in RNA splicing. However, available methods are not well suited for handling heterogeneous and large datasets. Such datasets scale to thousands of samples across dozens of experimental conditions, exhibit increased variability compared to biological replicates, and involve thousands of unannotated splice variants resulting in increased transcriptome complexity. We describe here a suite of algorithms and tools implemented in the MAJIQ v2 package to address challenges in detection, quantification, and visualization of splicing variations from such datasets. Using both large scale synthetic data and GTEx v8 as benchmark datasets, we assess the advantages of MAJIQ v2 compared to existing methods. We then apply MAJIQ v2 package to analyze differential splicing across 2,335 samples from 13 brain subregions, demonstrating its ability to offer insights into brain subregion-specific splicing regulation.

The usage of RNA sequencing (RNA-seq) has become ubiquitous in biomedical research. While some studies utilize RNA-seq only to investigate the overall expression level of genes, an increasing number of studies analyze changes in the relative abundance of gene isoforms. Changes in gene isoforms can occur through multiple mechanisms, including alternative promoter usage, alternative polyadenylation, and alternative splicing (AS). The production of different gene isoforms can in turn lead to diverse functional consequences, including changes to the translated protein domains, to degradation rates, and to localization. Previous studies showed that the majority of human genes are alternatively spliced with over a third of them shown to change their major isoform across 16 human tissues[1]. These observations, combined with the association of splicing defects with both monogenic and complex disease, serve to motivate the study of splicing variations across diverse experimental conditions. Consequently, independent labs as well as large consortia produce vast amounts of RNA-seq data. Datasets may involve anywhere from just a few to many thousands of samples each, and are typically heterogeneous as they often do not

represent biological or technical replicates. The consequent increased splicing variability, illustrated in Fig. 1a, b, can be the result of a multitude of factors, both experimental (e.g., difference in sequencing machine), and biological (e.g., sex, age). While some confounding factors may be corrected with appropriate methods[2], fully removing the observed variability in such data is unlikely and may also overconstrain the data, thus leading to a loss of true biological signal. Thus, there is a general need for methods that can effectively detect, quantify, and visualize splicing variations from large and heterogeneous RNA-seq datasets.

Broadly, the quantification of changes in gene isoform usage can be divided between methods that aim to quantify whole isoforms and those that quantify localized AS "events" within a gene. While quantifying all gene isoforms accurately across diverse conditions can be regarded as the grand challenge of transcriptomics, achieving this goal remains open due to several limiting factors. In the case of long reads technology, these factors include high error rate and high costs which do not allow researchers to capture enough reads from all isoforms. In

[1]Department of Genetics, University of Pennsylvania, Philadelphia, PA, USA. [2]Division of Human Genetics, Children's Hospital of Philadelphia, Philadelphia, PA, USA. [3]Department of Computer and Information Science, University of Pennsylvania, Philadelphia, PA, USA. [4]Institute for Translational Medicine and Therapeutics, University of Pennsylvania, Philadelphia, PA, USA. [5]These authors contributed equally: Jorge Vaquero-Garcia, Joseph K. Aicher, San Jewell, Matthew R. Gazzara. ✉e-mail: yosephb@upenn.edu

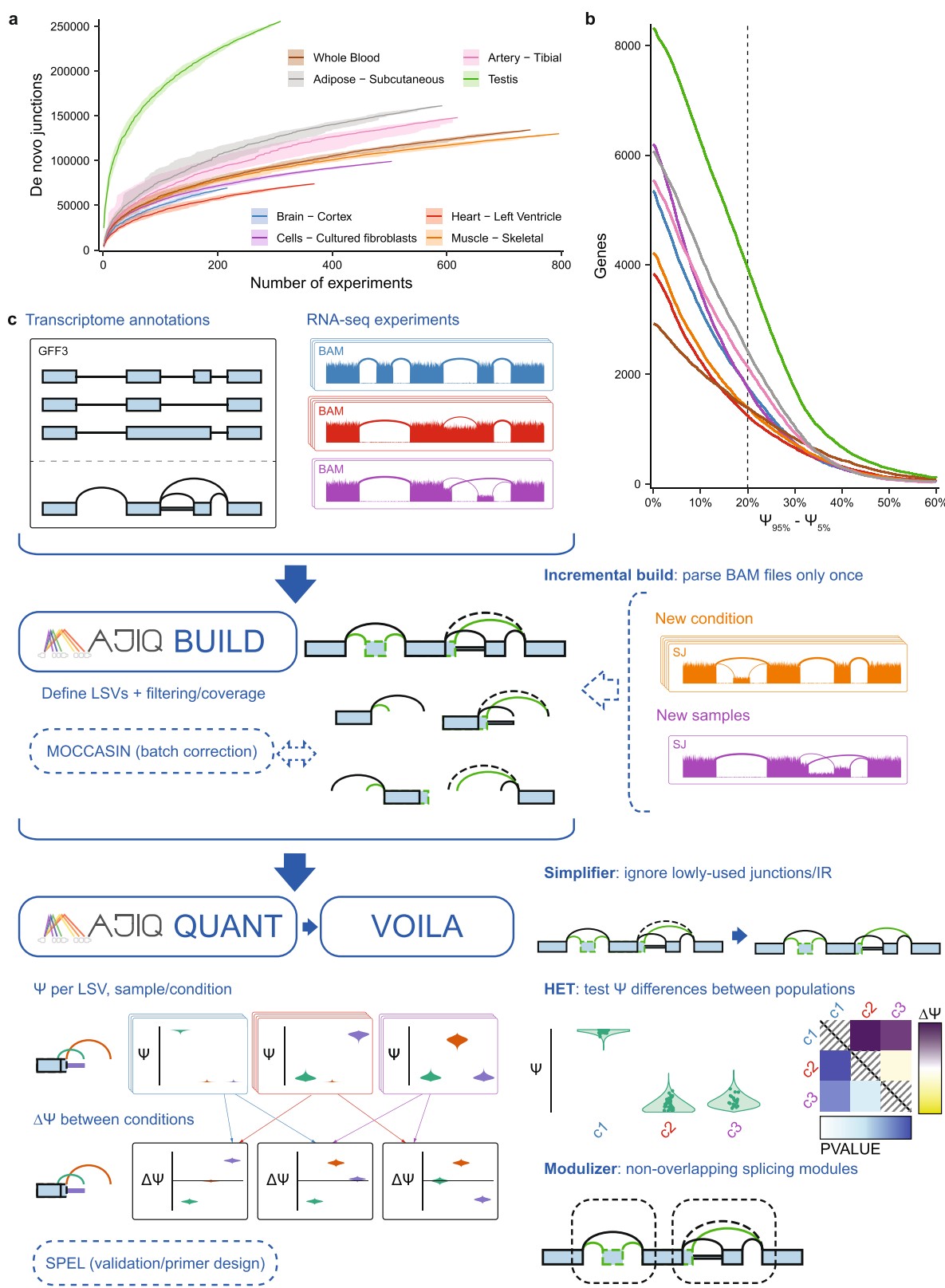

the case of the more commonly used short reads technology, these limiting factors include the sparsity of reads, their positional bias, and the fact that reads usually cannot be assigned to a unique isoform. In addition, the composition of isoforms in a sample is typically unknown, requiring further inference of the existing isoforms or making simplifying assumptions such as a known transcriptome. These issues have led many researchers to focus on local AS "events,"

which can be more easily and accurately quantified from RNA-seq. AS events are quantified in terms of percent spliced in (PSI, denoted by Ψ), which is the relative ratio of isoforms including a specific splicing junction or retained intron. Traditionally, AS events have been studied only for a restricted set of the most common "types" (e.g., cassette exons). In a previous study, we extended this set of AS event types using the formulation of local splicing variations (LSVs) and

**Fig. 1 | MAJIQ efficiently and accurately models, quantifies, and visualizes RNA splicing from large and complex RNA-seq datasets. a** The number of identified distinct unannotated de novo junctions increases with larger subsets of different tissues from GTEx. Lines show the median over 30 randomly selected permutations over experiments in each subset, confidence bands show the 5th to 95th percentiles over permutations of samples per tissue. **b** The number of genes with at least one junction where the difference between the 95th percentile and 5th percentile of PSI exceeds a given value for different tissues from GTEx (same tissues/colors as in **a**). Dashed vertical line indicates how many genes have a difference in PSI exceeding 20%. **c** MAJIQ combines annotated transcript databases and coverage from input RNA-seq experiments to build a model of each gene as a collection of exons connected by annotated and de novo junctions and retained introns (splicegraph). Junctions and retained introns sharing the same source or target exon form local

splicing variations (LSVs). MAJIQ quantifies the relative inclusion of junctions and retained introns in each LSV in terms of percent spliced in (PSI, Ψ) and provides VOILA to make interactive visualizations of splicing quantifications with respect to each gene's splicegraph and LSV structures. MAJIQ v2 introduces an incremental build, which allows RNA-seq coverage to be read from BAM files only once to a coverage file (SJ), accelerating subsequent builds with different experiments. MAJIQ v2 introduces a simplifier, which can be used to reduce splicegraph/LSV complexity by ignoring lowly used junctions and retained introns. MAJIQ v2 introduces a new mode for quantification, HET, which compares PSI differences between populations of independent RNA-seq experiments and accounts for variable uncertainty per experiment. MAJIQ v2 introduces the modulizer, which allows performing analysis relative to non-overlapping splicing modules rather than LSVs.

introduced MAJIQ as a software package for studying such LSVs[3]. LSVs, which can be defined as splits in a gene splicegraph coming into or from a reference exon, allow researchers to capture not only previously defined AS types, but also much more complex variations involving more than two alternative junctions (see examples in Fig. 1c for illustration). Furthermore, the LSV formulation, and similar definitions of local AS events suggested in subsequent works[4,5], also help incorporate and quantify unannotated (also termed de novo) splice junctions. Previous work comparing splicing across mouse tissues has shown that accounting for complex and de novo variations results in over 30% increase of detected differentially spliced events while maintaining the same level of reproducibility and experimental validation rates[3]. Importantly, capturing such unannotated splice variations is of particular importance for the study of disease such as cancer and neurodegeneration, which often involve aberrant splicing[6,7].

Despite previous demonstrations of MAJIQ's utility for analyzing AS[3,8], we found MAJIQ along with many commonly used methods for AS events quantification to be ill-suited for handling heterogeneous and large RNA-seq datasets. Such datasets pose several algorithmic, computational, and visualization challenges. First, the assumption of a shared PSI per LSV junction in a group, used by methods such as MAJIQ and LeafCutter, is violated in such data even when handling only a small dataset with few samples, leading to a potential increase in false positives and loss of power. Second, algorithms need to not only scale to thousands of samples efficiently but also to allow incrementally adding new samples as more data is acquired, and to support multiple group comparisons (e.g., multiple tissue comparisons across GTEx). Third, the increased complexity of the data requires efficient representation. Such efficient representation would allow users to capture the many unannotated splicing variations in the data, while at the same time simplifying its representation and quantification. Such simplification will allow to filter lowly used splice junctions while also detecting additional, non-classical sub-types of significant variations. Finally, efficient and user-friendly visualization is required to probe possibly multiple sample groups as well as individual samples.

To address the above challenges, we developed an array of tools and algorithms included in the MAJIQ v2 package. These include nonparametric statistical tests for differential splicing (MAJIQ HET), an incremental splicegraph builder, a new algorithm for quantifying intron retention, a method to detect high-confidence negative (non-changing) splicing events, and a new Modulizer algorithm to parse all LSVs across genes into modules which can then be classified into subtypes. These algorithms and tools are coupled with a new visualization package (VOILA v2), which allows users to compare multiple sample groups, simplify splicegraphs, and probe individual data points (e.g., LSV in an individual sample) while representing hundreds or thousands of samples. In addition, to support reproducibility, we develop a package for comparative evaluation of different methods for RNA splicing analysis and use it to demonstrate that the new version of MAJIQ compares favorably with the current state of the art using both synthetic (simulated) and real (GTEx) data. Finally, we apply the MAJIQ

v2 toolset to 2335 RNA-seq samples from 374 donors across 13 brain subregions. We use VOILA v2 to visualize the result and highlight several key findings in brain subregion-specific variations in cerebellar tissue groups compared to the remaining brain regions.

## Results
### The MAJIQ v2 splicing analysis pipeline
To support RNA splicing analysis using large RNA-seq datasets we implemented the set of tools and algorithms illustrated in Fig. 1c. In the first step, the MAJIQ builder combines transcript annotations and coverage from aligned RNA-seq experiments in order to build an updated splicegraph for each gene which includes de novo (unannotated) elements such as junctions, retained introns, and exons. Several user-defined filters can be applied at this stage to exclude junctions or retained introns which have low coverage or are not detected in enough samples in user-defined sample groups. Notably, per-experiment coverage is saved separately so that it can be used in subsequent analyses without reprocessing aligned reads a second time (i.e., incremental build). This feature is highly relevant for large studies with incremental releases, such as ENCODE and GTEx, and also for individual lab projects where datasets or samples are added as the project evolves.

In the second step of the pipeline, the MAJIQ quantifier is executed. As in the original MAJIQ framework, splicing quantification is performed in units of LSVs. Briefly, an LSV corresponds to a split in gene splicegraphs coming into or out of a reference exon. Each LSV edge, corresponding to a splice junction or intron retention, is quantified in terms of its relative inclusion (PSI, $\Psi \in [0, 1]$) or changes in its relative inclusion between two conditions (dPSI, $\Delta\Psi \in [-1, 1]$). Given the junction spanning reads observed in each LSV, MAJIQ's Bayesian model results in a posterior distributions over the (unknown) inclusion level ($\mathbb{P}(\Psi)$), or the changes in inclusion levels between conditions ($\mathbb{P}(\Delta\Psi)$). This model accounts not only for the total number of reads but also for factors such as read distribution across genomic locations and read stacks. Given its Bayesian framework, the model can also output the confidence in inclusion change of at least C ($\mathbb{P}(|\Delta\Psi| > C)$), or the expectation over the computed posterior distributions ($\mathbb{E}[\Psi]$, $\mathbb{E}[\Delta\Psi]$). In this work, we introduce two new algorithms within the MAJIQ quantifier. The first involves how intron retention is quantified, allowing for much faster execution with higher accuracy (see "Methods"). The second addition is the implementation of additional test statistics, termed MAJIQ HET (heterogeneous). Conceptually, the original MAJIQ model assumes a shared (hidden) PSI value for a given group of samples and accumulates evidence (reads) across these samples to infer PSI. In contrast, MAJIQ HET quantifies PSI for each sample separately and then applies robust rank-based test statistics (TNOM, InfoScore, or Mann–Whitney $U$). As we demonstrate below, the new HET test statistics allow MAJIQ to increase reproducibility in small heterogeneous datasets, and gain power in large heterogeneous datasets.

A new optional analysis step introduced here is the VOILA Modulizer, an algorithm which organizes all identified LSVs into AS

modules and then groups these modules by type. Briefly, AS modules represent distinct segments of a gene splicegraph involving overlapping LSVs which are contained between a single source and single target exon. However, unlike DiffSplice's AS modules[9], we do not use a recursive definition of these modules and instead classify all identified modules by their substructures into types. The module's substructures are in turn defined by the basic units of alternative splicing, namely intron retention, exon skipping and 3' or 5' splice variations. As we demonstrate below, the automatic AS module classification greatly facilitates a wide range of downstream analysis tasks.

The next step of the pipeline involves visualization of the quantified PSI and dPSI using VOILA v2. This new package runs as an app (on macOS, Windows, Linux) which supports the visualization of thousands of samples per LSV as violin beeswarm plots with multi-group comparisons and advanced user filters. Users can perform searches by gene name or junction, and simplify the visualization by filtering out lowly included junctions. This option is highly relevant for large heterogeneous datasets where many junctions might be captured but may not be relevant for specific comparisons/samples. Notably, unlike the builder filters described above, the VOILA v2 filters do not affect the underlying splicegraphs but only help declutter the visualization to aid in subsequent analysis. VOILA v2 has the option to run as a server to share results with collaborators while all of the pipeline's results can also be exported into other pipelines as tab-delimited files and for automated primer design for validation using MAJIQ-SPEL[10].

## Performance evaluation

In order to assess MAJIQ HET, our updated method for detecting differential splicing, we performed a comprehensive comparison to an array of commonly used algorithms using both synthetic and real data. We considered only algorithms capable of analyzing large datasets, including the original MAJIQ algorithm (upgraded with the v2 codebase to enable efficient data processing), rMATS turbo[11], LeafCutter[5], SUPPA2[12], and Whippet[4]. This analysis was performed using a 16 core machine and reports memory together with wall-time rather than CPU time to capture a more realistic estimate of running time on a modern desktop. However, since Whippet and SUPPA2 do not support multicore usage out of the box we implemented a user script for those methods that ran 16 jobs in parallel (denoted "(×16)"). Figure 2a shows the results of this analysis when running multi-group, multi-sample differential splicing task, typical for such datasets. In this case, we perform all pairwise comparisons between 10 tissue groups, and the number of samples in each group grows from 1 (10 total samples) to 6 (60 total samples). Supplementary Fig. 1 shows the results of this analysis without the addition of a parallelization script).

For MAJIQ V2, the builder step is roughly 6.6 times slower in wall-time compared to the quantification step, emphasizing the advantage of separating the two as the builder needs to only be executed once. All algorithms are able to process such large datasets using only 0.5–4 GB of memory, an amount readily available on modern laptops.

Nonetheless, large differences exist between the methods. Both SUPPA2 and Whippet are significantly faster than the other methods when using the additional parallelization script and much slower without it. Memory and time consumption also greatly depend on the task definition, with the biggest effect associated with read mapping. rMATS, LeafCutter and MAJIQ rely on aligned BAM files. Such files are commonly available for large datasets such as GTEX, or may be produced independently for other tasks such as expression analysis. In contrast, Whippet performs pseudo mapping internally which is computationally expensive, cannot be removed, and also limits it to annotated splice sites. However, when including STAR alignment time and memory into the analysis, or Salmon mapping for SUPPA2, Whippet exhibits significantly shorter running time (parallelized) and lower memory consumption. Finally, we note that even when comparing rMATS, LeafCutter and MAJIQ the computational tasks perform

by those methods are quite different. All three capture annotated "classical" splicing events, while MAJIQ and LeafCutter also capture complex events involving more than two alternative RNA segments, unannotated splice junctions and exon. In addition, MAJIQ is the only method that performs unannotated intron retention detection and quantification, a computationally expensive task.

Next, we assessed the accuracy of all algorithms using a large-scale synthetic dataset for comparing two tissue groups. This synthetic dataset, by far the largest of its kind to the best of our knowledge, was constructed to be "realistic" such that each synthetic sample was generated to mimic a real GTEx sample from either cerebellum or skeletal muscle tissues (see "Methods"). All methods were required to report changing AS events which pass the method's statistical significance test and inferred to exhibit a substantial splicing change of at least 20% (see "Methods"). However, we note that since the various algorithms use significantly different definitions of AS events it is hard to compare those directly. For example, LeafCutter defines AS events as clusters of overlapping introns which may involve multiple 3'/5' alternative splice sites and skipped exons, while rMATS is limited to only classical AS events with two alternative junctions. Thus, to facilitate a comparative analysis, we resorted to comparing the various algorithms output at the gene rather than event level using the synthetic dataset shown in Fig. 2b. A more refined analysis at the event level can be found in Supplementary material (Supplementary Fig. 2 and Supplementary Data 5–8) and exhibits similar trends to the ones reported here at the gene level. First, we found SUPPA2 consistently reported over 6000 differentially spliced genes, thousands more than any other method, while Whippet reported an average of 727 genes, significantly fewer than the other methods which reported over 2000 changing genes (Fig. 2b top bar chart). Whippet, followed by rMATS, reported significantly more non-changing events. Whippet, rMATS and SUPPA2 all exhibited high FDR ranging around 15–30%, as well as high average FNR of 32%, 60%, and 49%, respectively. Both MAJIQ and MAJIQ HET consistently maintained a lower false discovery rate compared to other algorithms (0.3%) and a low level of false negative rate, which was similar to that of LeafCutter. On small sets, for example when using 5 samples per group, LeafCutter had a slightly lower FNR (2.5% vs 5.5% for HET), but MAJIQ exhibited slightly lower FDR (0.03% vs 0.8%). However, we note that unlike regular classification tasks, the aforementioned statistics are not computed over a fixed set, as each method reports on a different set of genes. For example, while both MAJIQ and LeafCutter achieve similarly high Matthew Correlation Coefficient (MCC) statistic of 0.97–0.99 across groups sizes (Fig. 2b, bottom), MAJIQ reports overall 34% more genes as changing (2337 vs 1739) and 6% more as non-changing (7110 vs 6713) compared to LeafCutter (Fig. 2b, top). These differences are further amplified when considering changes at the event rather than gene level: 4267 events reported as changing by MAJIQ versus 2169 by LeafCutter (see Supplementary Data 5–8). This increased difference is mainly due to the increased resolution of event definition by MAJIQ. Specifically, MAJIQ uses the local splice variations formulation described above, while LeafCutter uses a definition of overlapping intronic regions which give rise to coarser event definition that can be sensitive to the coverage threshold used.

The significant differences between the methods described above raises the question how the reported sets of differentially spliced genes overlap. Figure 2c illustrates the result of such analysis when using 10 samples per group. Here, we looked at the intersection between different methods at the gene level and when a set was unique to a method (i.e., the underlying events are well defined) we also estimated the associated FPR. We found SUPPA2 reports a significantly higher number of unique genes (1713) as differentially spliced but over a quarter of those are false positives. The next set sizes are those for Whippet (356), LeafCutter (324), HET and SUPPA2 (248), HET (230), and MAJIQ HET and DPSI (181) with a FPR of 9.5% for the LeafCutter's

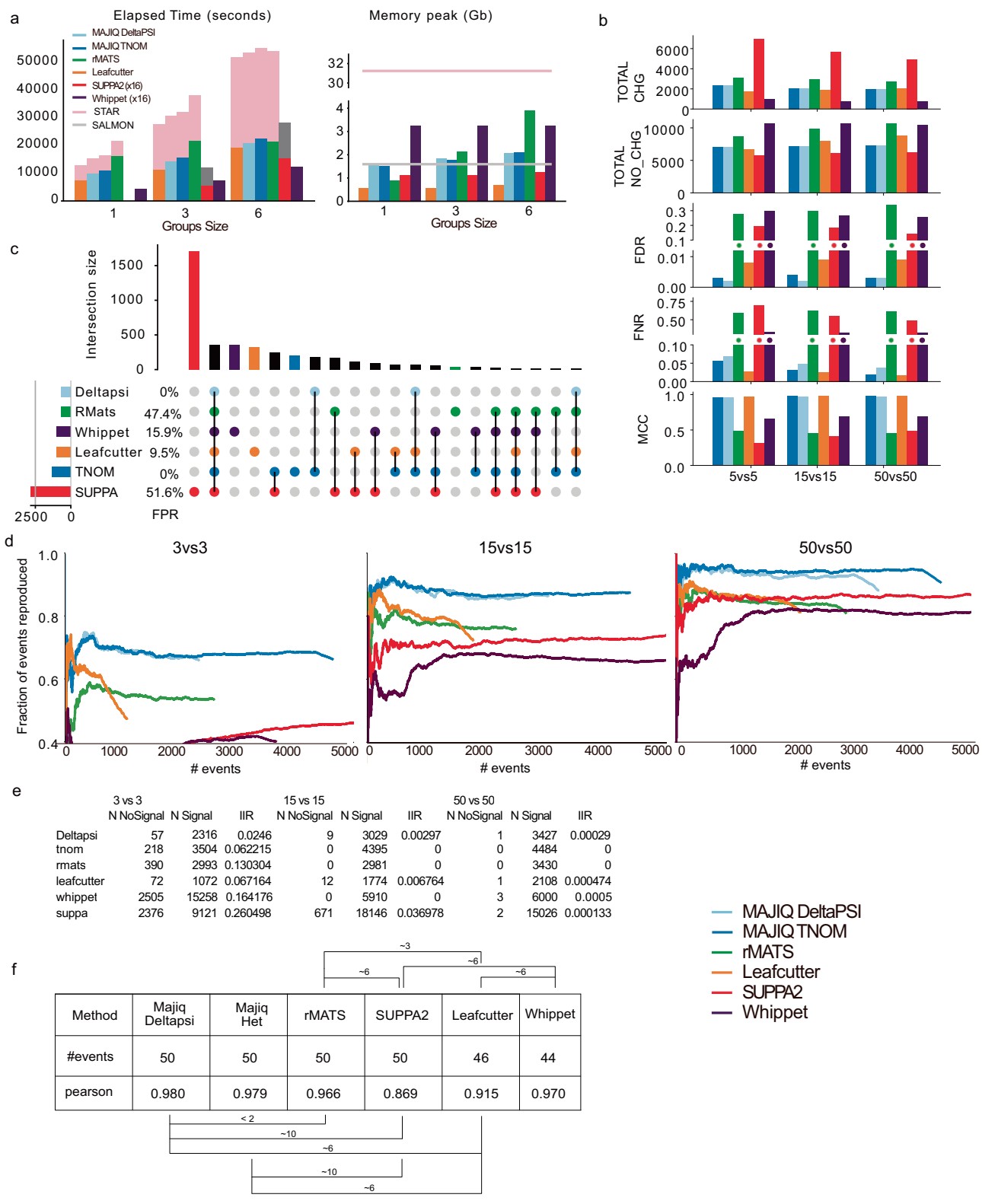

unique set and close to 0 FPR for both MAJIQ's algorithms unique sets. rMATS and Whippet report significantly fewer unique genes with a high false positive rate of 47% and 16%, respectively.

Next, we turned to assess performance on real GTEx data using several metrics. Here, unlike the synthetic data analysis which focused on comparative evaluation at the gene level, we focus on the actual AS events reported by each method. First, we used the reproducibility ratio (RR) statistic as shown in Fig. 2d. The RR plots follow a similar procedure to that of irreproducible discovery rate (IDR) plots, used

extensively to evaluate ChIP-seq peak callers[3,13]. Briefly, RR plots answer the following simple question: given an algorithm $A$ and a dataset $D$, if we rank all the events that algorithm $A$ identifies as differentially spliced $(1, ..., N_A)$, how many would be reproduced if you repeat this with dataset $D'$, comprised of similar experiments using biological or technical replicates? The RR($n$) plot, as shown in Fig. 2d, is the fraction of those events that are reproduced ($y$-axis) as a function of $n \le N_A$ ($x$-axis), with the overall reproducibility of differentially spliced events expressed as RR($N_A$) (far right point of each curve in

**Fig. 2 | Performance evaluation using synthetic and real data. a** Time (left) and memory (right) consumption for running all pairwise differential splicing analysis between 10 GTEx tissue groups, with number of samples per group increasing from 1 to 6 (*x*-axis). The "x(16)" label denotes a parallelization script added to methods not supporting multithreading. **b** Performance evaluation, aggregated over genes, for differential splicing calls using simulated GTEx samples (cerebellum and skeletal muscle). Metrics include the total number of genes reported as changing (TOTAL-CHG) or non-changing (TOTAL-NO-CHG), with the resulting FDR, FNR, and Matthew's correlation coefficient (MCC). Horizontal axis denotes set size. **c** Upset plot based on the 10vs10 analysis shown in **b**. The bars on top represent the overlap between genes reported as differentially spliced by each method indicated below. The bars and FPR values by each method's name refer to genes reported only by that method. **d** Reproducibility ratio (RR) plots for real data, using GTEx cerebellum and muscle samples. Plots are based on each method's reported list of splicing events (not genes) and unique scoring approach. *X*-axis is the ranked number of

events reported and *Y*-axis is the fraction of those events reproduced within the same number of top-ranking events when repeating the analysis using a different set of samples from the same tissues. Line length represents the total number of differentially spliced events reported (see "Methods" for details). RR graphs are shown for group sizes of 3 (left), 15 (middle), and 50 (right). **e** Intra-to-Inter Ratio (IIR) results for GTEx samples as in **d**. IIR is the ratio between the number of events reported as significantly changing when comparing two sample groups of the same type ("NoSignal" column) and the number of events reported as significantly changing when comparing groups of different types (muscle versus cerebellum, as in **d**). **f** Correlation between estimated dPSI and RT-PCR quantification of splicing changes. RT-PCR results taken from ref. [3] based on mouse liver and cerebellum RNA, extracted by ref. [46] for the matching RNA-Seq samples. All splicing events examined are annotated cassette exons. Two-sided *p* values based on the Dunn and Clark's *z* procedure estimated via ref. [47].

Fig. [2]d). In our RR analysis using groups of size 3–50 GTEx samples each, we found both MAJIQ and MAJIQ HET compared favorably to the other methods, but with the new HET algorithm exhibiting improved detection power resulting in a higher number of AS events at the same reproducibility level. These results were robust with respect to the specific random subset of samples selected (Supplementary Fig. 3, top).

The second statistic we used for evaluating performance on real data is the intra-to-inter ratio (IIR)[8], which serves as a proxy for FDR on real data where the labels are unknown. Specifically, IIR computes the ratio between the number of differentially spliced events reported when comparing groups of the same condition (e.g., brain) and the number of events reported for similar group sizes of different conditions (e.g., brain vs liver). In our work, we found IIR to be a lower bound estimate of true FDR, though it lacks theoretical guarantees. In the analysis shown in Fig. [2]e, we found IIR to behave similarly to FDR on synthetic data with MAJIQ, MAJIQ HET, and LeafCutter exhibiting low IIR of 2–7% even for small group sets of 5 samples, while rMATS, SUPPA2, and Whippet had an IIR of 13%, 26% and 23.7% respectively. However, unlike FDR on synthetic data, IIR dropped much more significantly, hitting practically zero for all methods for large sample groups. This result is to be expected since the IIR statistic compares sample groups of the same type, unlike the synthetic dataset described above where different tissues are compared. These results were robust to the specific random subset of samples selected (Supplementary Fig. 3, bottom).

The last component we included for the methods assessment is a comparison of each methods' dPSI accuracy irrespective of the specific statistical test used or ranking. On the synthetic data described above we plotted a CDF of absolute deviation between computed and actual dPSI for all cases reported to exhibit a change of at least 20% or 10% (Supplementary Fig. 4). We found that MAJIQ, Whippet and LeafCutter all performed generally well, while rMATS and SUPPA exhibited larger deviations. MAJIQ-HET compared favorably to all other methods, especially when the number of samples per group was low and when using a more permissive threshold of 10%. Finally we assessed PSI quantification accuracy by comparing it to triplicates of RT-PCR assays, the gold standard in the RNA field. Using over 100 such experiments from two different mouse tissues we previously produced[3,8], we computed the Pearson correlation between RT-PCR and each method's quantification, along with the statistical significance of observed differences (Fig. [2]f). Overall, we found MAJIQ, Whippet and rMATS all achieving high correlation of approximately 0.97–0.98, while Leaf-Cutter and SUPPA had significantly lower correlation of 0.915 and 0.869, respectively (see Supplementary Fig. 5 for scatter plots per method). We note that this analysis for LeafCutter was possible since all events we tested were simple cassette exon skipping, but it is not clear how to translate LeafCutter's output to actual PSI in the general case.

## VOILA v2 enables visualization of thousands of samples

To facilitate visualization and downstream analysis of both the new outputs from MAJIQ HET over large, heterogeneous datasets and traditional MAJIQ PSI or MAJIQ dPSI quantification over replicate experiments, we developed VOILA v2 as a server based cross-platform app. Replacing the previous HTML file based visualization with VOILA v2 allows for interactive visualization of all LSVs in all genes, with data ranging from one sample to thousands of samples. After an initial indexing step that is run one time, users can now, on the fly, filter their data by several criteria including dPSI levels between groups, read coverage over junctions, LSV types and complexity, and the statistical test for significance, as opposed to re-running VOILA with the filtering criteria, as was required in the previous version. Another advantage of the new VOILA v2 is its ability to run as a server so that results can be shared with collaborators without the need to transfer large files.

To highlight these new features, we ran MAJIQ HET and VOILA v2 on GTEx v8 brain tissues which are known to exhibit high levels of alternative splicing. Overall, this analysis involved 2335 RNA-seq samples from 374 donors across 13 tissue groups (see "Methods"). Figure [3] shows the VOILA view for this large dataset for the key splicing factor gene *PTBP1*, including a splicegraph (top) with combined read information from 225 cerebellum RNA-seq samples. Users can easily add and remove splicegraphs for other tissue groups or individual samples of interest. Figure [3] bottom panel shows a VOILA visualization for quantifying a single junction in a single LSV across the 2335 RNA-seq samples. Here, the 13 tissues are displayed as violin beeswarm plots with each point representing a single sample which can be interrogated by hovering the user's cursor over it. Finally, VOILA uses a heatmap (Fig. [3] bottom right) to represent the pairwise differences between the tissue groups for the junction of interest. The upper half of the heatmap represents the difference in medians of $\mathbb{E}[\Psi]$ distributions between the tissue groups, while the bottom half represents the *p* values associated with these group differences (see "Methods"). For the example LSV and junction in *PTBP1*, the cerebellar tissues (cerebellum and cerebellar hemisphere) show a distinct splicing pattern with reduced usage of this junction (lower $\mathbb{E}[\Psi]$ values in the left-most violin plots) which was significant according to MAJIQ HET (Mann–Whitney *U* shown) (Fig. [3]).

## VOILA Modulizer defines alternative splicing modules to facilitate downstream analysis

The LSV and junction showcased in the above example are of biological importance. PTBP1 is a widely expressed splicing factor that binds CU-rich sequences, but it is downregulated during neurogenesis which contributes to neuronal splicing patterns[14–16]. Decreased activity of PTBP1 in neuronal tissues is attributed to numerous mechanisms, some of which involve splicing regulation of two cassette exons in the region highlighted in the *PTBP1* splicegraph (Figs. [3] and [4]a boxed regions)[3,17], making differences between brain

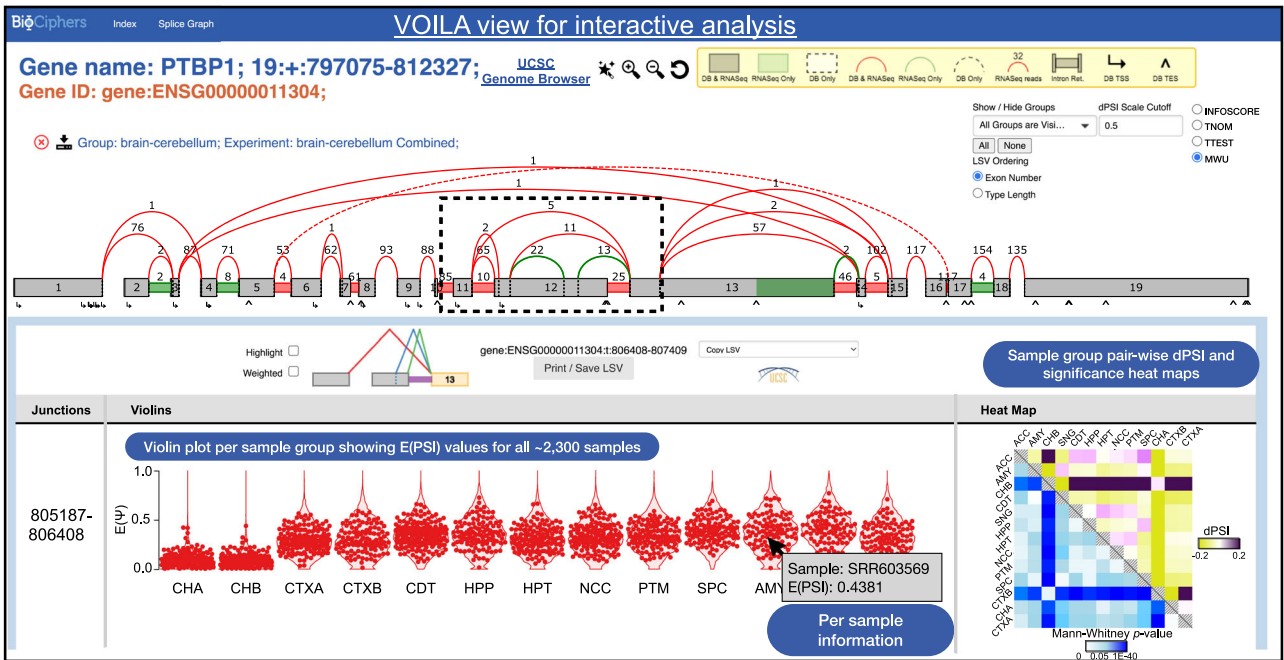

**Fig. 3 | Enhanced visualization of large datasets with VOILA v2.** VOILA view of MAJIQ HET output for 13 brain tissue groups from GTEx from 2335 RNA-seq samples originating from 374 unique donors. Top portion shows gene information and filtering criteria as well as the splicegraph for *PTBP1* showing median read counts from 225 cerebellum samples. Bottom portion displays visualization and PSI quantification for each junction in each LSV for the gene of interest. Here the distribution of $\mathbb{E}[\Psi]$ values across the indicated tissue groups (abbreviations given in Fig. 5a) is displayed as a violin beeswarm plot for the red junction for the exon 13 target LSV, represented in the cartoon, for all 2335 RNA-seq samples. Individual sample information is given by hovering the cursor over individual points that represent each sample (gray box). Bottom right heatmap displays MAJIQ HET quantifications of all group pairwise comparisons across the 13 brain tissue groups to highlight significant splicing changes. Yellow to purple color scale on the top right indicates the expected $\Delta\Psi$ between tissue groups while blue color scale on the bottom left indicates the significance of the difference between group PSI distributions for one of four statistics used by MAJIQ HET (Mann–Whitney displayed).

subregions of potential interest. Mammalian-specific, neuronal skipping of an alternative cassette exon in the linker region between the second and third RNA recognition motifs (RRMs) of *PTBP1* (exon 12 in the splicegraph) results in a protein isoform of PTBP1 with reduced repressive activity leading to altered splicing patterns during neuronal differentiation[17]. Additionally, in mouse brain we previously described inclusion of a unannotated, premature termination codon (PTC) containing cassette exon with conserved splice sites in humans that shows increased inclusion in mouse cerebellum (compared to brainstem and hypothalamus) and is developmentally regulated through murine cortex development[3]. While LeafCutter analysis of *PTBP1* on all of GTEx failed to detect this event in human tissues, we find evidence of de novo splice junction reads corresponding to both the conserved 3′ and 5′ splice sites of this unannotated exon that we validated previously in mouse (Fig. 4a), suggesting this exon is also included in human brain tissues.

This region of the splicegraph is complex, however, and is defined by overlapping LSVs each with multiple splice junctions and intron retention detected (Fig. 4a: exon 11 source LSV, left; exon 13 target LSV, right). While the LSV formulation has several benefits, including accurate PSI quantification of complex splicing patterns involving more than two splice junctions[3], it is difficult for users to know which junction quantifications and combinations of junctions from different LSVs should be combined to define common alternative splicing (AS) events, like the cassette exons described above in *PTBP1*. Moreover, while certain annotated and de novo junctions may have sufficient read coverage for detection and quantification by MAJIQ, they can be very lowly included in a user's condition(s) of interest. For example, several hundred reads across GTEx brain samples support the existence of the annotated, intron distal alternative 3′ss of exon 12 of *PTBP1*, but source LSV quantification of the relative usage of this junction is low across all samples (Fig. 4a, left. Blue junction median PSI across samples of <5% in all tissue groups). Such junctions add additional complexity to the splicegraph and may hinder definition of common AS event types across the transcriptome.

To overcome these limitations and to facilitate downstream, transcriptome wide analysis of common AS event types we developed the VOILA Modulizer (Fig. 4b). First, users have the option to simplify the splicegraph to remove junctions that do not meet a threshold for raw read coverage, low inclusion levels across the input samples ($\mathbb{E}[\Psi]$), and/or low relative splicing changes between input comparisons between sample groups ($\mathbb{E}[\Delta\Psi]$) (Fig. 4bi). This helps remove junctions that do not meet a user's desired threshold for biological significance and facilitate downstream event definitions, like the alternative 3′ss of exon 12 of *PTBP1* discussed above with low inclusion levels across all sample groups (blue junction in Fig. 4a, left). Next, the simplified splicegraph is traversed to define single entry, single exit regions of the splicegraph that we call alternative splicing modules (AS modules or ASMs), as shown for part of *PTBP1* (Fig. 4bii). Within each AS module, pattern matching is performed between the remaining exon and junction structure of the simplified splicegraph to each of 14 basic AS event types (Supplementary Fig. 8a). This process is illustrated in Fig. 4biii for two AS modules within *PTBP1*. We note that this step can lead to some redundant event information (e.g., intron retention events sharing the same junction and intron coordinates, as in Fig. 4b). Because these events are quantified from both sides through a source and a target LSV, the quantification in terms of PSI or dPSI between conditions may not agree and thus both are provided. Nonetheless, downstream filtering can ensure agreement when counting event types and defining changing events.

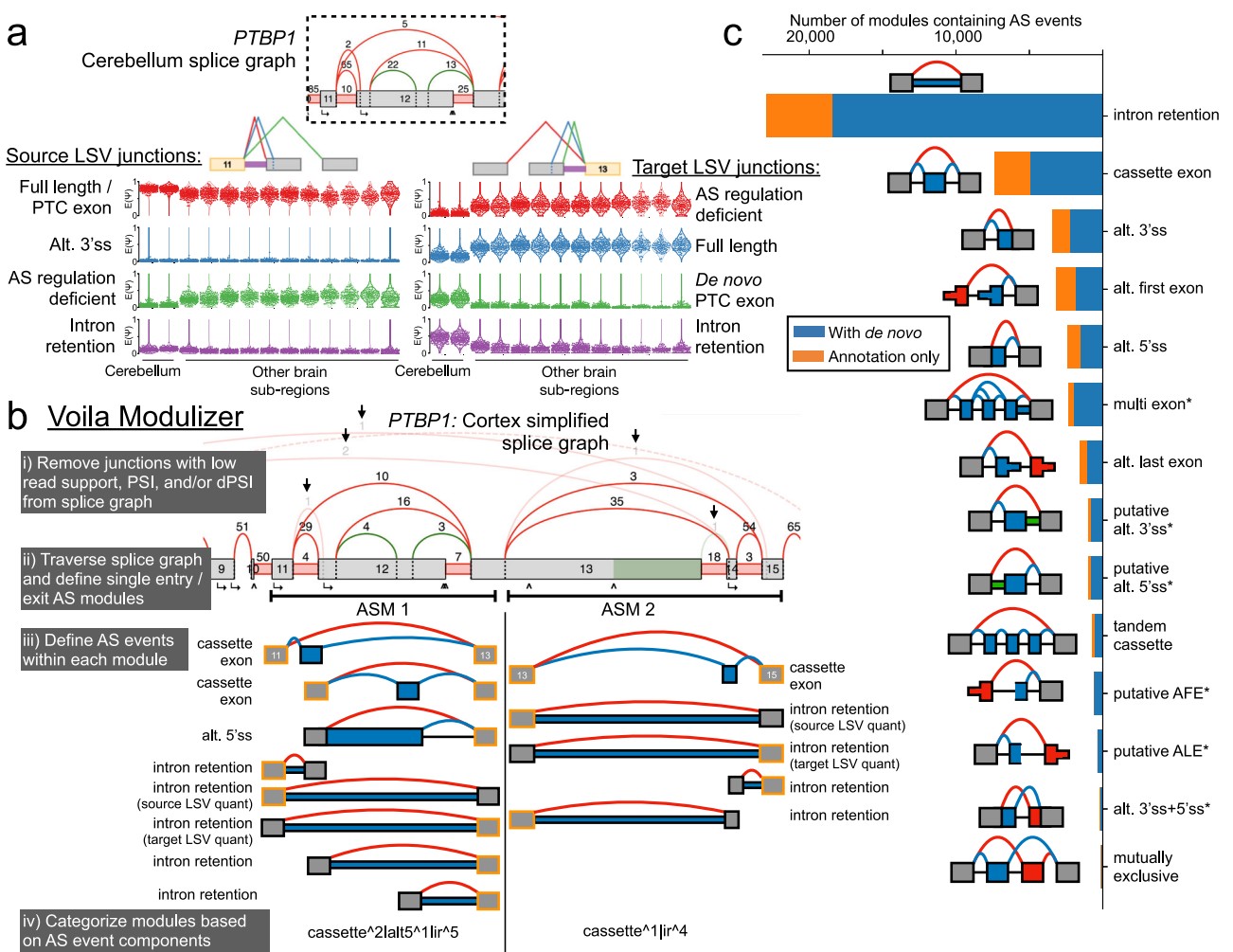

**Fig. 4 | Downstream analysis of alternative splicing modules with VOILA v2.**
**a** Top shows region of human *PTBP1* splicegraph (with reads from combined cerebellum samples) and two LSVs corresponding to a mammalian specific exon skipping event that alters PTBP1 splicing regulatory activity[17] (green junction in exon 11 source LSV, left; red junction in exon 13 target LSV, right) and de novo detection of a conserved, PTC-containing exon previously shown to be included in mouse neuronal tissues[3] (green junction in exon 13 target LSV). Bottom shows distribution of PSI across the 13 brain tissue groups as well as annotation of each junction. **b** VOILA Modulizer workflow (gray boxes) and an example region of the *PTBP1* splicegraph where junctions that did not meet a median $\mathbb{E}[\Psi]$ value of 5% or more in any of the 13 brain tissue groups were removed (arrows). Two alternative splicing modules (ASMs) were defined as single entry, single exit regions of the splicegraph and within these modules binary, AS events are defined. Gray exons

highlighted in yellow indicate reference exons that belonged to LSVs for which MAJIQ quantification exists. Blue junctions and exonic or intronic regions indicate inclusion of the alternative region of the event and red junctions indicate exclusion of the alternative region. **c** Stacked bar chart showing the number of binary AS event types that make up AS modules across the 13 brain tissue groups from GTEx. AS event types are represented with a cartoon to the left of the chart and are named to the right of bars. Asterisks indicate non-classical AS event types. Each junction or intron had to have a median of $\mathbb{E}[\Psi]$ values of 5% or more across the samples of at least one tissue group to contribute to AS module definitions. Blue regions indicate AS events that contained de novo junctions and/or introns not found in the annotated transcripts (Ensembl v94) while orange regions indicate AS events containing only annotated junctions and introns.

Running the VOILA Modulizer produces a number of files based on event types with a uniform structure containing coordinate and quantification for each sample group to facilitate downstream analysis on AS modules and AS event types of interest (Supplementary Fig. 8a). Some AS event definitions identified by the Modulizer are analogous to those defined by other splicing quantification algorithms that only handle binary, classical splicing events (e.g., MISO[18] or rMATS[11]). However, the MAJIQ + VOILA Modulizer approach adds a number of benefits compared to other available algorithms. First, our approach allows for de novo splice junction and intron retention detection, which is crucial in the context of GTEx brain subregions. Using a simplification threshold of median $\mathbb{E}[\Psi]$ over brain tissue groups of ≥5% to be included in the simplified splicegraph, we defined 32,435 AS modules where 70.6% contain at least one unannotated splice junction and/or intron retention (Fig. 4c and Supplementary Fig. 8b). The AS module

formulation also allows for definition of common splicing patterns across brain subregions beyond binary splicing events, which made up 59.2% of all AS modules. The remaining 40% of AS modules contained multiple AS events which, in many cases, involved mixing of a classical event type with intron retention (Supplementary Fig. 8b). Both at the AS event level (Fig. 4c) and at the AS module level (Supplementary Fig. 8b), intron retention was particularly common using our simplification threshold of median $\mathbb{E}[\Psi]$ of greater than 5% in any one brain tissue group. This is consistent with previous studies that have found certain neuronal tissues to have very high levels of intron retention compared to other contexts[19].

Initial analysis of the most common AS module types led us to add additional splicing event patterns to our definitions, beyond those that are classically defined in other tools (intron retention, cassette exon, alternative 3' and 5'ss, alternative first and last exons, tandem cassette

exons, and mutually exclusive exons[11,18]). These included putative alternative first and last exons, where at least one alternative exon is created from a de novo junction that does not belong to any nearby exon, and putative alternative 3′ or 5′ss, where a cassette exon has an inclusion junction removed during simplification (low inclusion) with sufficiently high intron retention levels (see Supplementary Fig. 8a for full details). Taken together, these additional, non-classical splicing event types participated in the make up of 13.1% of AS modules or 11.6% of AS events overall detected in the brain (Fig. 4c, event types marked with asterisks). Importantly, the Modulizer outputs all of these event types in a format amenable to downstream regulatory analysis, which will facilitate the future characterization of these splicing patterns (Supplementary Fig. 8a).

To expand on the finding of widespread intron retention and how this alters AS events and AS module definitions beyond neuronal tissues, we analyzed a subset of GTEx samples from each of the 53 tissues (see "Methods"). We observed a range of expected Percent Intron Retention (PIR) across all tissues where both cerebellar tissues have a relatively higher degree of intron retention, when compared to other tissues (Supplementary Fig. 9a, b). Surprisingly, the other neuronal tissues ranked towards the bottom of all GTEx tissues along with heart and muscle (Supplementary Fig. 9a, b). To see how these differences in intron retention may affect the definition of AS events and AS modules, we selected a subset of 9 tissues across the range of intron retention levels (Supplementary Fig. 9b, arrowheads) and ran the VOILA Modulizer. We used an $\mathbb{E}[\Psi] \geq 5\%$ simplification threshold on each of these single tissues individually or on all 53 GTEx tissues together and compared these results to the brain subregion analysis above. At an AS event level, intron retention was the most common event detected within AS modules where it was detected in over 75% of modules from single tissues with high levels of IR (e.g., cerebellum or spleen) down to as little as 53% of modules in the tissue with the lowest level of IR (skeletal muscle, Supplementary Fig. 9c). Other AS event types were similarly detected across single tissues or tissue groups, with notable exceptions like a larger fraction of modules containing cassette exons in skeletal muscle or a higher fraction of AS modules containing all other AS event types when all 53 GTEx tissues were analyzed together (Supplementary Fig. 9c). Similarly, at the AS module level, including all 53 tissues in the VOILA Modulizer analysis as a group led to the definition of more complex AS modules where 41.8% of the detected modules contained two or more unique AS event types, when compared to the brain tissue group alone (Supplementary Fig. 9d). Notably, analysis of the of the nine single tissues also led to a significant portion of between 25 and 30% of modules containing two or more unique AS event types (Supplementary Fig. 9d). Analyzing the frequency of the top ten AS module types that were defined by the brain tissue analysis (defined in Supplementary Fig. 8b) showed similar trends where most AS module types were stably detected across diverse tissue types with diverse levels of intron retention (Supplementary Fig. 9e). In all, this analysis highlights the importance of considering more complex splicing patterns through the use of MAJIQ with the VOILA Modulizer, even when analyzing a single condition.

## Analysis of unique cerebellar splicing patterns highlights regulatory programs

Finally, we wished to use MAJIQ + VOILA Modulizer to analyze differential splicing patterns between brain subregions. Previous studies focused on splicing quantitative trait loci within GTEx brain tissues found the cerebellar tissues cluster separately from other brain subregions based on splicing[20]. Our analysis of *PTBP1* (Fig. 4a) and pairwise analysis of the number of significant LSVs according to MAJIQ HET further supports distinct splicing patterns in cerebellar tissues (Supplementary Fig. 10a). For these reasons, we sought to identify AS modules and events with unique splicing patterns in the cerebellum. Using the above AS module definitions from all junctions and introns

with group level median $\mathbb{E}[\Psi] \geq 5\%$, we next searched for consistent splicing changes between the two cerebellar tissues (cerebellum and cerebellar hemisphere) and other brain subregions using MAJIQ HET. We required an absolute difference in median $\mathbb{E}[\Psi]$ values of 20% or more when comparing both cerebellar tissue groups to the same other brain region tissue group in addition to having a Mann−Whitney two-sided $p < 0.05$ (Fig. 5a, see "Methods").

From these comparisons we found 3995 unique, changing AS modules (Fig. 5b) comprising over 7500 changing AS events (Supplementary Fig. 10b). At the changing AS module and AS event levels, intron retention was most prevalent, followed by cassette exons and other mixtures of binary AS event types with intron retention (Fig. 5b). As with the analysis based on inclusion levels alone (Fig. 4), most changing AS modules (53.3%) consisted of multiple, binary AS event types (Fig. 5b), highlighting the prevalence of complex splicing changes and the power of MAJIQ + VOILA Modulizer approach.

Alternative splicing regulation of cassette exons in neuronal tissues is very well studied with a number of expression changes associated with splicing factors (e.g., expression of the RBFOX family, down regulation of PTB proteins, expression of NOVA proteins, etc.)[15,16]. For this reason we wished to analyze the regulatory signature around the cassette exons defined from our MAJIQ HET + VOILA Modulizer analysis to see if we could capture known, and potentially novel, regulatory motifs around cerebellar cassette exons.

Our initial analysis focused on all changing cassette exon (CE) events. This mirrors the CE landscape that would be identified by other, event-based splicing quantification algorithms and consists of a combination of CEs which come from modules consisting of only a single CE event (Fig. 5b, c brown dot) in addition to those from complex modules with multiple event types (Fig. 5b, c purple dots). Because RNA binding proteins bind short motifs and splicing factor binding that results in alternative splicing regulation typically occurs proximal to the splice sites of an alternative exon[21], we performed a Z-score analysis for hexamer occurrence within 300 nucleotides upstream or downstream of cerebellar changing cassette exons versus those alternative exons that did not change between brain subregions (see "Methods"). Moreover, because splicing factors typically act in position-specific manners (e.g., binding downstream of a cassette exon enhances exon inclusion while binding upstream represses inclusion)[22,23], we further separated cassette events into those with increased exon inclusion in cerebellar tissues (Fig. 6a, blue) and those with increased exon exclusion in cerebellar tissues (Fig. 6a, red) when compared to other brain subregions.

Supporting the validity of our approach, this analysis uncovered a number of motifs either upstream or downstream of cerebellar cassette exons with known links to neuronal splicing regulation. For example, for cerebellar inclusion cassettes we found a number of CU-rich and UGC containing hexamers upstream and the RBFOX-binding-motif, UGCAUG[24], enriched downstream (Fig. 6a, blue). SRRM4/nSR100 is known to bind UGC-containing sequences upstream of neuronal microexons to enhance their inclusion with the aid of SRSF11 that binds CU-repeat sequences[25]. Accordingly, motif maps across our different cerebellar exon classes based on hexamers shown to bind SRRM4[26] and SRSF11[25] by iCLIP show clear enrichment of these motifs just upstream of cerebellar inclusion cassette exons (Supplementary Fig. 11a, c). This result is consistent with increased expression of these two genes in cerebellar tissues leading to enhanced intronic splicing enhancer (ISE) activity around these events (Supplementary Fig. 11a−e).

In addition to SRRM4 and SRSF11, the RBFOX family is highly expressed in neuronal tissues and is known to enhance exon inclusion when it binds downstream of the 5′ss[27,28]. Indeed, we find a strong enrichment of the known UGCAUG-binding site just downstream of cerebellar inclusion events (Fig. 6a and Supplementary Fig. 11f, blue). This result is consistent with increased expression of these genes and

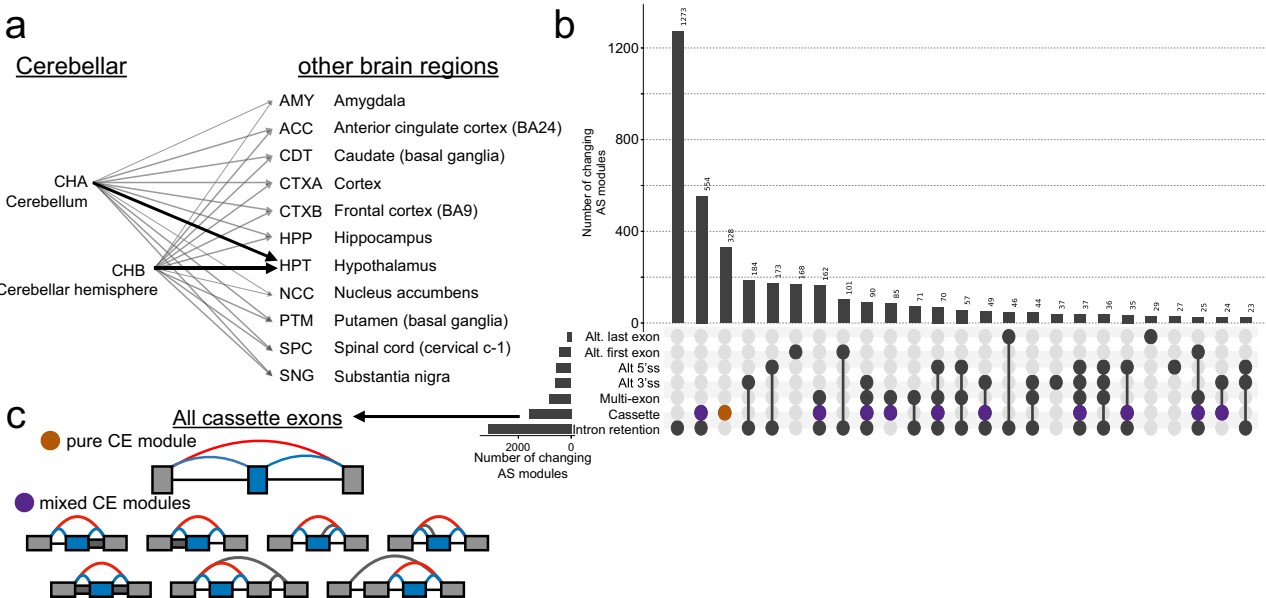

**Fig. 5 | MAJIQ HET + VOILA Modulizer defines the complex landscape of cerebellar splicing changes. a** Pairwise comparisons run through MAJIQ HET to find significant splicing changes between GTEx cerebellar tissues (cerebellum and cerebellar hemisphere) versus the other 11 brain tissue groups. Dark arrows indicate an example of a consistent change, where both cerebellar tissue groups versus the same other brain region, the hypothalamus, shared a significant change. Alternative splicing (AS) modules were kept for downstream analysis if at least one such consistent comparison was significant (see "Methods"). GTEx abbreviations are given for each tissue. **b** Upset plot showing the consistent, significantly changing AS event type(s) that make up AS modules. AS events had to have an absolute difference in median $\mathbb{E}[\Psi]$ of 20% or more when comparing both cerebellar tissue groups (cerebellum and cerebellar hemisphere) to the same other brain region tissue group in addition to having a Mann–Whitney two-sided $p < 0.05$ as reported by MAJIQ HET. **c** Examples of cassette exon (CE) types with consistent changes between cerebellar and other brain tissues. All CEs have quantified inclusion junctions (blue junctions) and a shared exclusion junction (red junction), potentially within a mixture of other AS event types (gray junctions and introns) (purple circles on upset plot in **b**).

increased ISE activity in the cerebellum versus other brain subregions (Supplementary Fig. 11h, i).

Because RBFOX proteins were previously shown to regulate alternative splicing by binding distal intronic regions (>500 nucleotides from a splice site)[29], we also computed Z-scores for 6-mers in the distal intronic regions upstream and downstream of cerebellar inclusion versus non-changing exons. We found the RBFOX hexamer was among the motifs with the highest Z-scores in the downstream distal intronic region, although to a lesser degree compared to UGCAUG in downstream proximal intronic region (distal $Z = 5.38$, proximal $Z = 21.08$). The top motifs for distal intronic regions both upstream and downstream of cerebellar inclusion exons were CA-repeats (Supplementary Fig. 11g), which are the known binding sites for HNRNPL and HNRNPLL[30]. While both L and LL are expressed in the brain, with HNRNPLL showing tissue-enriched protein expression in the cerebellum in GTEx samples[31], they have not, to our knowledge, been linked to distal intronic splicing regulation in the brain.

Interestingly, we found hexamers containing motifs known to bind QKI (e.g., ACUAA containing[32]) were enriched around both cerebellar inclusion (upstream) and exclusion events (downstream) (Fig. 6a). QKI is known to act as a splicing enhancer when it binds downstream of cassette exons and represses exonic inclusion when it binds upstream[32]. We generated a motif map of the QKI hexamer (ACUAAY[33]) around these exon classes and found clear positional enrichment proximal to the regulated splice sites in both exon sets (Fig. 6b, top). Moreover, we generated RNA maps of in vivo binding events (determined by CLIP peaks) of QKI across multiple cell types and found enriched binding consistent with the motif maps (Fig. 6b, bottom, Supplementary Fig. 11j). Compared to other brain subregions, the two cerebellar tissues exhibited lowest expression of QKI (Fig. 6c). This result points to a regulatory mechanism by which decreased expression of QKI in cerebellum may contribute to both cerebellar

exon exclusion events (loss of enhancing activity downstream leading to exon skipping) and cerebellar exon inclusion events (loss of repressive activity upstream leading to inclusion) (Fig. 6d). To further support this model, we analyzed RNA-seq from ENCODE where QKI was knocked down in HepG2 cells via shRNA and searched for evidence of QKI regulation of the junctions involved in cerebellar cassette exon events ($|\Delta\Psi| \geq 20$ with Mann–Whitney two-sided $p < 0.05$ upon knockdown, see "Methods"). We found cerebellar cassette exon events were overall 5.6 times more likely to show evidence of QKI regulation when compared to non-changing events (two-tailed binomial $p < 2.8 \times 10^{-31}$). Importantly, when we considered the direction of splicing change upon QKI depletion, we found a significant enrichment of both cerebellar inclusion and exclusion events that were consistent with our model (i.e., QKI depletion promotes cerebellar splicing patterns, fold-enrichment versus non-changing >4.2, two-tailed binomial $p < 6.4 \times 10^{-11}$, Supplementary Fig. 11k). We did not observe a statistically significant enrichment of cerebellar inclusion or exclusion events that were inconsistent with our model (fold-enrichment versus non-changing <1.3, two-tailed binomial $p > 0.44$, Supplementary Fig. 11k).

Given that many regulated cassette exons occur within AS modules containing other AS event types (Fig. 5b, c), we next wished to explore if regulatory motifs differed between these subsets. Because AS modules containing cassette exons and those containing both cassette exon and intron retention events are common (Fig. 5b and Supplementary Figs. 8b and 9e), we chose to stratify the set of all cassette exons into those that contained a regulated intron retention event and those that occurred in AS modules in which intron retention was not detected. We calculated Z-scores for hexamers from these exon subsets by comparing them against the set of exons that were not changing in cerebellar comparisons and compared the results of the two analyses. Figure 6e shows an example of this analysis for hexamers located downstream of cerebellar exclusion cassette exon subsets. The

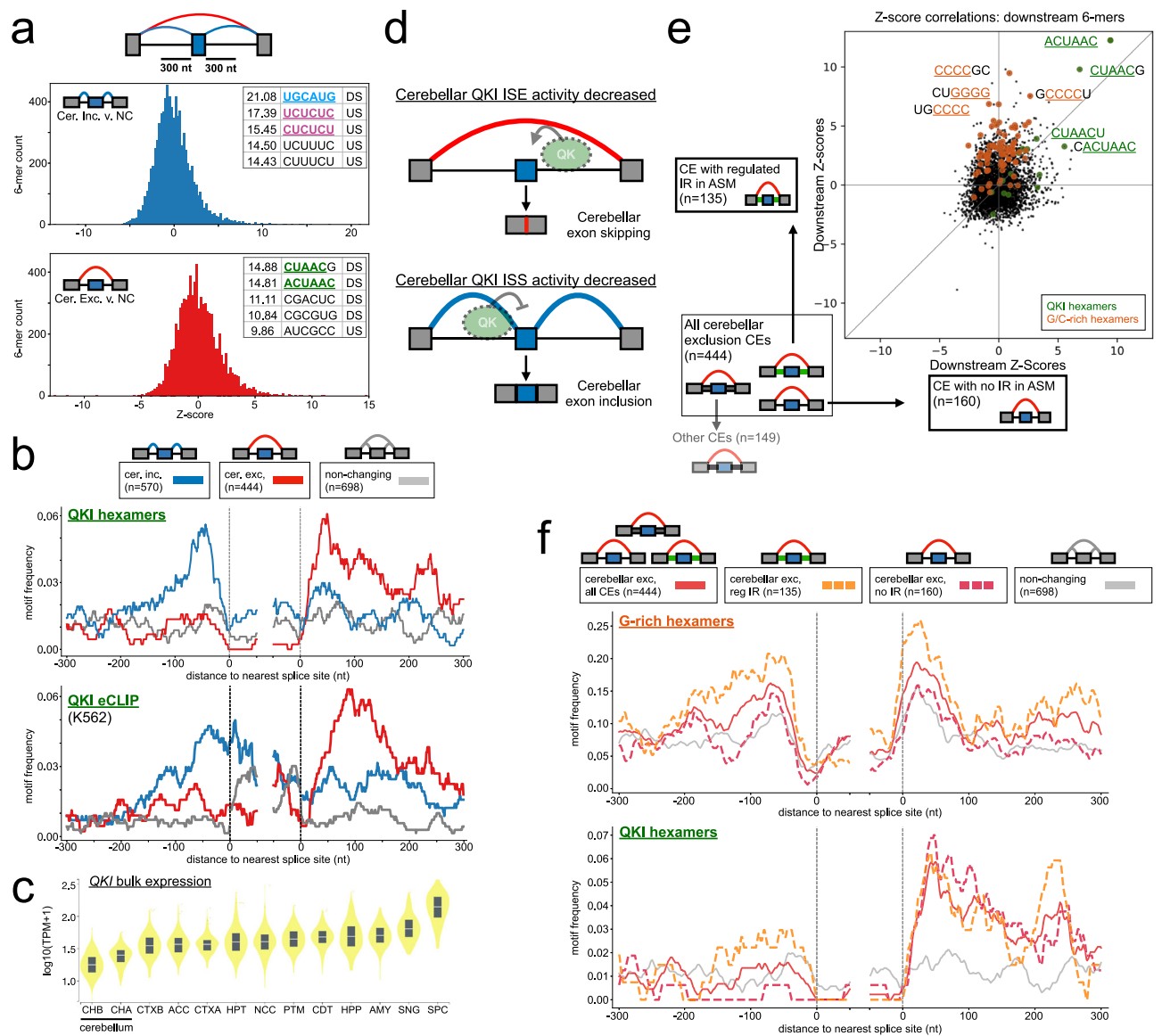

**Fig. 6 | Regulatory analysis of simple and complex cerebellar cassette exon types. a** Distribution of hexamer *Z*-scores within 300 nucleotides upstream or downstream of all CE event types (Fig. 5c) for cerebellar inclusion versus non-changing (top, blue) or cerebellar exclusion versus non-changing (bottom, red) (see "Methods"). Top motifs for RBPs of interest are highlighted (QKI (green), RBFOX (blue), SRRM4 (yellow), SRSF11 or PTB (purple)). All motifs and *Z*-scores are given in Supplementary Data 1. **b** RNAmaps showing the frequency of QKI hexamers (ACUAAY, top) or binding of QKI (K562 eCLIP peaks, bottom) around cerebellar inclusion (blue), exclusion (red), or non-changing (gray) CEs. Frequencies calculated over windows of 20 nucleotides smoothed by a running mean of 5 nucleotides. **c** GTEx *QKI* brain tissue expression ($\log_{10}(1 + TPM)$) generated using gtexportal.org. Violins represent the distribution of values, boxes represent the 25th and 75th percentiles, white lines represent medians, and outliers beyond 1.5 times the interquartile range are shown. Each tissue is represented by no fewer than

109 samples. Abbreviations are defined in Fig. 5a. **d** Model for QKI position-dependent regulation in GTEx brain tissues. Decreased expression of QKI in cerebellar tissues results in decreased downstream intronic splicing enhancer (ISE) activity, resulting in cerebellar exon exclusion (top). Decreased upstream intronic splicing silencer (ISS) activity results in cerebellar exon exclusion (bottom). **e** Scatter plot showing hexamer *Z*-score correspondence for non-overlapping sets of cerebellar CE exclusion events: (*y*-axis) CE exclusion events from AS modules containing changing intron retention (IR) event(s) versus non-changing and (*x*-axis) CE exclusion events from AS modules without IR event(s) detected. Motifs of interest are highlighted according to colors in the inset. **f** RNAmaps (plotted as in **b**) for given cerebellar CEs stratified by intron status for G-rich hexamers (five of six positions are G and contains GGGG, top) or QKI hexamers (ACUAAY, bottom). Red lines, all cerebellar exclusion CEs; orange dashed, subset of exclusion CEs with changing IR; fuchsia dashed, subset of exclusion CEs with no IR; gray, all non-changing CEs.

top two hexamers that match QKI binding motifs (ACUAAC and CUAACG) found when analyzing all CE events (Figs. 5c and 6a) also had the highest *Z*-scores in the intron retention regulated and no intron retention CE subsets (Fig. 6e, green). On the other hand, several of the G- and C-rich motifs that were enriched downstream of all CE cerebellar exclusion events (Fig. 6a and Supplementary Data 1) were biased towards higher *Z*-scores solely in the CE subset that contained regulated intron retention (Fig. 6e, orange). This is consistent with

observations from previous studies analyzing intron retention events that found retained introns tended to be more G/C-rich when compared to non-retained introns[19]. Motif maps across the different cerebellar exclusion CE sets supported the *Z*-score analysis and highlight that the enrichment of G-rich sequences (Fig. 6f, top) and C-rich sequences (Supplementary Fig. 12a, b) around all cerebellar exclusion CEs is driven mostly by the subset of CEs containing a regulated intron retention event (compare dashed orange and dashed fuchsia lines).

The QKI hexamer showed similar positional enrichment downstream of both CE subsets (Fig. 6f, bottom).

Similar results were seen when comparing Z-scores for upstream and downstream hexamers identified in the all cassette exon analysis (Figs. 5c and 6a) of cerebellar inclusion and exclusion CE subsets stratified by intron status (Supplementary Fig. 12a). While some of the motifs found in the complete CE analysis scored similarly in subsets stratified by intron retention status (e.g., the RBFOX hexamer or SRRM4 hexamers around cerebellar inclusion exons), others showed biased enrichment in CEs with regulated intron retention compared those with no intron retention (e.g., CU-repeat hexamers) (Supplementary Fig. 12). Overall, this analysis highlights some shared and distinct regulatory features of cerebellar cassette exons with and without evidence of intron retention.

## Discussion

The work presented here represents the culmination of continuous development of MAJIQ since its original release in 2016[3]. The original MAJIQ, like many other algorithms, was designed for comparing relatively small groups of RNA-seq experiments from biological replicates. However, as we demonstrate here using GTEx v8, datasets nowadays can easily grow to hundreds and thousands of non-replicate samples. The sheer size and heterogeneous nature of such data poses challenges related to efficiency, ability to capture but also simplify de novo and complex splicing variations, ability to identify event subtypes, and the ability to visualize such events and subtypes. To address these challenges we developed MAJIQ v2 with algorithmic improvements as well as the simplifier, the modulizer, incremental build options, and the VOILA v2 visualization package. In addition, we perform extensive comparison of MAJIQ v2 to other algorithms, create a resource for reproducible algorithm comparison, and demonstrate the utility of MAJIQ v2 in a detailed splicing analysis of more than 2300 samples from GTEx v8 brain subregions.

With respect to performance, we showed MAJIQ v2 compares favorably to available methods in terms of efficiency, accuracy on synthetic data, and reproducibility on real RNA-seq data. When comparing MAJIQ HET to MAJIQ dPSI from ref. [3] that was ran with the new v2 code base, we found both exhibited similar reproducibility, but HET offered a significant increase in detection power. Finally, in terms of efficiency, we found MAJIQ v2 performed similarly to the most efficient tools in both memory and time. This is a notable achievement given that MAJIQ is the only tool amongst those that offers detection and quantification of de novo intron retention, a computationally expensive yet important task as we discuss below.

The extensive evaluations performed here are accompanied with data and code which is aimed to serve as resources for the community. Specifically, we created the largest synthetic RNA-seq dataset to date, with over 300 samples. This data was generated based on real life GTEx samples, quantified by RSEM, rather than MAJIQ modeling assumptions. A related contribution is the evaluation package we created, validations-tools. This package allows users to not only reproduce our results but also to easily add other tools or specific datasets of interest and repeat the analysis. We recommend researchers and cores to take advantage of this as it is possible that on a dataset with other characteristics the various algorithms would perform differently. We hope the data and code provided here will help avoid software misuse and lack of reproducibility that previously affected the assessment of MAJIQ and other splicing software[34]. More generally, the reproducibility tools we included here should help future developers to achieve at least the "bronze" level of reproducibility as was recently proposed[35].

Finally, applying our improved pipelines to GTEx brain subregions allowed us to map the complex alternative splicing patterns observed across over 2300 heterogeneous human neuronal tissue samples from 374 donors and 13 tissue groups. Our approach and subsequent analysis offers several advances compared to previous efforts such as

refs. [5,20] for mapping brain sub-region splicing patterns. First, we offer improved quantification accuracy and the ability to capture de novo and complex splicing events as well as retained introns (IR). Furthermore, as we illustrated for cerebellum-specific regulation, the definition of AS modules and AS event types we introduce here greatly facilitates downstream analysis. Specifically, we were able to find regulatory signatures of known neuronal cassette exon splicing programs (i.e., the RBFOX family, SRRM4 with SRSF11, PTBP1, and QKI[15,16,25,36]) and to discover additional regulatory complexity between the cassette exons subsets that contain or lack intron retention events. We anticipate the ability introduced here to interrogate AS modules and their components will facilitate future regulatory discoveries in other datasets from additional biological contexts.

We note that there are key limitations to the regulatory analysis we performed for cerebellar-specific splicing, which was based solely on bulk tissue RNA-seq experiments from GTEx. Previous work leveraging single cell data to deconvolute bulk GTEx tissues into their relative cell type compositions suggests that cerebellar tissues contain relatively larger proportions of neurons compared to other brain subregions[37]. This fact can confound the interpretation of our results in terms of neurobiology as neurons are known to express certain splicing factors (e.g., RBFOX3/NeuN, SRRM4), which may explain the cerebellar splicing pattern we observed here. Thus, future directions for improving MAJIQ involve accounting for cell-type heterogeneity as well as combining long reads for isoform specific deconvolution. Other promising directions for future exploration include analysis of RNA sequencing for clinical diagnostics and exploiting MAJIQ's advantages for improved sQTL analysis.

In summary, we introduced here a significant update to the original MAJIQ package. We hope the analysis we performed, along with the tool, data, and evaluation package we supply, will inspire many more researchers to delve into splicing regulatory analysis in their own data and make exciting discoveries.

## Methods
### MAJIQ builder

In this subsection, we review how the MAJIQ builder prepares the structure and observations per experiment that are used for downstream splicing quantification as part of a scalable and principled approach to splicing analysis of large numbers of experiments. We describe the MAJIQ builder's new approach for estimating intron read rates, which allows junction and intron coverage to be calculated once and reused efficiently for multiple analyses, unlike other methods that quantify intron retention. We also describe the MAJIQ simplifier, which reduces the complexity of the structural models of splicing used in quantification that especially arises from the analysis of large and heterogeneous datasets.

MAJIQ encodes the set of all possible splicing changes for a gene in terms of a splicegraph. A splicegraph is a graph-theoretic representation of a gene's splicing decisions from one exon to another, with exons as vertices and junctions and retained introns as distinct edges connecting exons. The exons of each gene are non-overlapping genomic intervals. Each junction has a source and target exon with a position within each exon, indicating the positions that are spliced together when the junction is used. Retained introns are between adjacent exons and indicate that intron retention between the exons is possible.

MAJIQ first constructs each gene's splicegraph by parsing transcript annotations from a GFF3 file. Exon boundaries and junctions from each transcript for a gene are combined in order to produce the minimal splicegraph that includes each transcript's annotated exons and junctions, splitting exons by retained introns to ensure that each junction starts and ends in different exons. MAJIQ then updates the splicegraph with de novo junctions and introns found from processing input RNA-seq experiments' junction and intron coverage.

MAJIQ processes aligned input RNA-seq experiments to per-position junction and intron coverage in the following way. First, MAJIQ identifies reads with split alignments. The genomic coordinates of each split corresponds to a potential junction. Meanwhile, the coordinate of the split on the aligned read is the junction's "position" on the read. MAJIQ counts the number of reads for each junction from each possible position. Afterwards, MAJIQ identifies reads that contiguously intersect known or potential introns (i.e., reads that intersect the genomic coordinates between adjacent exons without splits within the intron boundaries). If the intron start is contained in the aligned read, the intron "position" is defined as for junctions (treating the exon/intron boundary as a junction with zero length). For aligned reads intersecting the intron but not the start, additional positions are defined by the genomic distances of the first positions of the aligned reads to the intron start. These additional positions per intron increase the number of ways aligned reads can intersect introns in comparison to junctions. To adjust for this and model intron read coverage similarly to junction read counts, MAJIQ aggregates together adjacent intron positions to the equivalent number of possible positions per junction, taking the mean number of reads per reduced positions.

MAJIQ uses the obtained junction and intron coverage to update the splicegraph in the following way. Each potential junction is mapped to matching genes by prioritizing (1) genes that already contain the junction (i.e., annotated junctions) over (2) genes where both junction coordinates are within 400 bp of an exon, which are prioritized over (3) genes where the junction is contained within the gene boundaries. The input experiments are divided into user-defined build groups. MAJIQ adds a de novo junction to the splicegraph if there is sufficient evidence for its inclusion in one of the build groups. This happens when the total number of reads and total number of positions with at least one read exceeds the user-defined minimum number of reads and positions in at least a minimum number of experiments. MAJIQ adds new de novo exons or adjusts existing exon boundaries to accommodate the added de novo junctions as previously described. Potential introns are added to the splicegraph under similar criteria, and their boundaries are adjusted or split to accommodate the adjusted or de novo exon boundaries.

Since processed intron coverage is averaged over the entire original intronic region, we can carry over the same coverage as an estimate for all resulting splicegraph introns, which are contained in the original intron's boundaries. In contrast, MAJIQ's previous approach, which is also used by most other tools that quantify intron retention, quantified intron coverage using local counts of unsplit reads sharing the position of known junctions. These local counts must be calculated using information from all processed experiments (for all de novo junctions), which requires samples to be reprocessed each time an analysis with different samples are performed. MAJIQ's new approach allows intron coverage to be processed once and used for multiple builds with potentially different intron boundaries. This enables MAJIQ's new incremental build feature, which saves intermediate files with junction and intron coverage that can be calculated once and reused instead of BAM files for multiple builds. This reduces storage and time processing experiments that are part of multiple analyses.

While MAJIQ uses raw totals of read rates and number of nonzero positions for adding junctions and introns to the splicegraph, the MAJIQ builder performs additional modeling of per-position read rates for use in quantification. First, we mask positions with zero coverage and with outlier coverage. Outlier coverage is assessed under the observation that per-position read rates generally follow a Poisson distribution. For each junction/position, we use all other positions with nonzero coverage for that junction to estimate the Poisson rate parameter. Then, MAJIQ calls any position with an extreme right-tailed $p$ value (default $10^{-7}$) under this model an outlier and ignores its contribution to coverage for quantification. Second, we perform bootstrap sampling of the total read rate over unmasked positions in order to model measurement error of true read rates. Under the assumption that each unmasked position is identically distributed, MAJIQ performs nonparametric sampling with replacement to draw from a distribution with identical mean and variance as the observed positions (see Supplementary Note). Since we assume that our read rates are generally overdispersed relative to the Poisson distribution, MAJIQ replaces nonparametric sampling with Poisson sampling when the nonparametric estimate of variance is less than the mean (i.e., underdispersed).

MAJIQ performs quantification of splicing events modeled as LSVs, which are defined by a splicegraph. A source (target) LSV is defined for an exon as a choice over the incoming (outgoing) edges to (from) that exon from (to) a different exon. In general, only LSVs with at least two edges are considered. MAJIQ builder prepares output files with raw and bootstrapped coverage for each junction/intron in each LSV for quick use by downstream quantifiers.

We observed that builds from many build groups or with high coverage tend to have increasingly complex splicegraphs and LSVs with many junctions. Many of these junctions are often lowly used in all the samples but were included in the splicegraph because they had enough raw reads and positions (noisy de novo) or are part of an unused annotated transcript. This motivated the MAJIQ simplifier, which allows junctions and introns to be masked from the final splicegraph used for quantification. After the splicegraph is constructed using all input build groups, MAJIQ calculates the ratio of the raw read rate for each junction/intron relative to the other junctions/introns in each LSV. If a junction has consistently low coverage in each of the build groups relative to the other choices in the two LSVs it can belong to, it is "simplified" and removed from the final splicegraph. This reduces the complexity of the final splicegraph and quantified LSVs, making output files smaller and downstream quantification more efficient.

In summary, the MAJIQ builder combines transcript annotations and input RNA-seq experiments in order to build a splicegraph encoding all possible splicing events consistent with both annotations and data and to prepare read coverage for quantification in terms of LSVs. The MAJIQ builder's updated approach for estimating intron read rates allows junction and intron coverage to be calculated once and reused as part of an incremental build for multiple analyses, unlike other methods that quantify intron retention. The MAJIQ builder also introduces an approach for simplifying the complexity that arises in splicing events when processing large numbers of experiments. Overall, this allows the MAJIQ builder to produce structural models of possible splicing events and read coverage for downstream quantification that scale to the setting of large numbers of RNA-seq experiments.

### MAJIQ quantifiers

MAJIQ provides three methods for quantifying RNA-seq experiments. MAJIQ PSI, MAJIQ dPSI, and MAJIQ HET, which we introduce in this paper. MAJIQ PSI and dPSI, which were previously described in ref. [3], quantify groups of experiments that are assumed to be replicates with a shared true value of PSI per group. MAJIQ PSI estimates a posterior distribution of PSI ($\Psi$) for a single group, while MAJIQ dPSI compares these distributions for two groups in order to estimate a posterior distribution for dPSI ($\Delta\Psi$). MAJIQ HET compares two groups of samples but drops the replicate experiments assumption, enabling analysis of more heterogeneous samples. Instead, experiments are quantified individually and groups are compared under the assumption that the true values of PSI are identically distributed between the two groups.

All three pipelines share the same underlying machinery for inferring posterior distributions for $\Psi$. Formally, $\Psi$ for a junction in an LSV is defined as the fraction of expressed isoforms using the junction out of all expressed isoforms containing the LSV. This fraction is not directly observable. Instead, we observe the number of reads aligned $r_j$

to each junction $j$ in the LSV. We model each $r_j$ as a realization of a binomial distribution over the isoforms with probability $\Psi_j$:

$$r_j \sim \text{Binomial}\left(\sum_{j \in \text{LSV}} r_j, \Psi_j\right). \tag{1}$$

We take a Bayesian approach to integrate prior knowledge of $\Psi$, allowing for improved estimation when there is low read coverage. This requires a prior distribution on $\Psi$. We previously observed that most values of $\Psi$ are nearly zero or one, which can be modeled using a generalization of the Jeffrey's prior for an LSV with $J$ junctions:

$$\Psi_j \sim \text{Beta}\left(\frac{1}{J}, 1 - \frac{1}{J}\right). \tag{2}$$

This prior is conjugate to the binomial likelihood, allowing for efficient closed-form estimation of the posterior distribution of $\Psi_j$ given the observed number of reads:

$$\Psi_j | \left\{r'_j : j' \in \text{LSV}\right\} \sim \text{Beta}\left(\frac{1}{J} + r_j, 1 - \frac{1}{J} + \sum_{j' \neq j} r'_j\right). \tag{3}$$

Since MAJIQ build obtains bootstrap replicates of observed read rates, we perform this posterior inference on each set of bootstrap replicate read rates to obtain an ensemble of posterior distributions.

For MAJIQ PSI, we obtain this ensemble of posteriors for replicate experiments by adding the observed read rates from the experiments that pass more stringent reads and position thresholds than the builder. MAJIQ PSI treats the average of the posterior distributions as a final distribution over $\Psi$. It reports point estimates of $\Psi$ as the mean of this distribution ($\mathbb{E}[\Psi]$) and saves a discretized version of the distribution for visualization in VOILA.

MAJIQ dPSI takes this a step further by using the posterior distributions on $\Psi_1$, $\Psi_2$ for two groups in order to compute $\Delta\Psi = \Psi_2 - \Psi_1$ between the two groups. We start by computing the distribution of $\Delta\Psi$ under the assumption of independence of $\Psi_1$ and $\Psi_2$ by marginalizing the product of their distributions:

$$\mathbb{P}_{\text{ind}}(\Delta\Psi) = \sum_{\Psi_2 - \Psi_1 = \Delta\Psi} \mathbb{P}(\Psi)_1 \mathbb{P}(\Psi)_2. \tag{4}$$

We know that $\Psi_1$ and $\Psi_2$ are not independent, so we integrate our knowledge that $\Delta\Psi$ is usually close to zero as a prior on $\Delta\Psi$. Following our previous work, we formulate our prior $\mathbb{P}_{\text{prior}}(\Delta\Psi)$ as a mixture of three components: (1) a spike around $\Delta\Psi = 0$, (2) a broader centered distribution around $\Delta\Psi = 0$, and (3) a uniform slab. We determine our final posterior distribution on $\Delta\Psi$ by adjusting $\mathbb{P}_{\text{ind}}(\Delta\Psi)$ by the prior and renormalizing:

$$\mathbb{P}(\Delta\Psi) \propto \mathbb{P}_{\text{ind}}(\Delta\Psi)\mathbb{P}_{\text{prior}}(\Delta\Psi). \tag{5}$$

MAJIQ dPSI computes point estimates of $\Delta\Psi$ using the posterior mean of the distribution ($\mathbb{E}[\Delta\Psi]$) and identifies confidence of measured changes in inclusion as posterior probabilities $\mathbb{P}(|\Delta\Psi| > C)$.

MAJIQ HET takes a different approach for comparing inclusion between two groups of experiments. MAJIQ HET drops the assumption of replicate experiments to consider heterogeneity in $\Psi$ between experiments within a group. Instead, MAJIQ HET assumes that the values of $\Psi$ per experiment in each of the groups come from the same distribution. We evaluate this assumption using null hypothesis significance testing. Null hypothesis significance testing is performed using one (or more) of four tests: (1) Welch's two-sample $t$ test, (2) Mann–Whitney $U$-test, (3) Total Number of Mistakes (TNOM) test, and (4) InfoScore test. Welch's two-sample $t$ test and Mann–Whitney $U$-test

are well-documented elsewhere[38,39]. Our implementation of Mann–Whitney $U$-test computes exact $p$ values when there are at most 64 experiments and computes asymptotic $p$ values using normal approximation with tie and continuity correction for larger samples. Meanwhile, the InfoScore and TNOM tests are adapted from ScoreGenes[40]. The TNOM test evaluates how well a single threshold on PSI can discriminate between the observed values in the two groups. The Total Number of Mistakes is the minimum number of misclassified observations under the best possible thresholds. The distribution on TNOM when the distributions are equal are calculated using the closed-form formula in ref. [41] to obtain $p$ values. Similarly, the Info-Score test evaluates how well a single threshold discriminates between groups, but, instead of measuring misclassifications directly, it identifies the threshold with the highest mutual information between the threshold and the true group labels. MAJIQ HET uses the dynamic programming algorithm in ref. [41] to evaluate the distribution of Info-Score under the null hypothesis in order to obtain $p$ values. All four tests require observed values of $\Psi$ per experiment, which is not directly observed. MAJIQ HET accounts for variable uncertainty per experiment in our estimations of $\Psi$ by repeated sampling of $\Psi$ from the posterior distributions of quantified samples. MAJIQ HET computes the $p$ value for each repeated sample of $\Psi$ over quantified experiments and reports the 95th percentile over the resulting $p$ values. These $p$ value quantiles are not calibrated, so MAJIQ HET also computes $p$ values with the posterior means of $\Psi$. MAJIQ HET also reports the median of the observed posterior means of $\Psi$ for each group. These $p$ values and the difference between the median observed posterior means are used together downstream in VOILA for the identification of high-confidence differentially spliced LSVs.

## VOILA

VOILA provides a suite of post-processing and visualization tools designed to allow researchers to make use of MAJIQ quantifications directly, or easily format and filter the output for passing to other post-processing tools.

The VOILA viewer acts as a complete visualization tool for interactive analysis of output from MAJIQ PSI, dPSI, or HET. It includes search and filter mode for all discovered LSVs, as well as an in-depth viewer for the full splicegraph of a gene and all of the LSVs found within it. When using the VOILA viewer with output from MAJIQ HET, VOILA will also automatically generate heatmaps for each LSV with the to quickly indicate the discovered $\Delta\Psi$ and statistical results from each group comparison. The viewer front end runs completely within a web browser interface, so it is able to function with similar results on any modern operating system without installation of special frameworks or system libraries. The viewer can also be configured to run as a standalone web server such that the interactive results can be easily shared with collaborators. Tutorials and parameters are made available to integrate VOILA with a wide range of common web server production software.

VOILA also has a number of modes for filtering and rearranging data into a number of human and machine-readable files. Determining confidently non-changing (background) and confidently changing events is one of the primary use cases. We define highly-confident non-changing events from MAJIQ HET as being (1) above a nominal $p$ value threshold, (2) within-group variance is sufficiently low as measured by IQR, and (3) between-group $\Delta\Psi$ is sufficiently low as measured by difference in medians. We accept that the between-group $\Delta\Psi$ threshold may be redundant in combination with the other two thresholds. We define confident changing events from MAJIQ HET as being (1) below a $p$ value threshold and (2) between-group $\Delta\Psi$ is sufficiently high as measured by difference in medians.

In addition to the basic text output modes, there is a separate comprehensive output mode dedicated to finding specific event types/patterns called the VOILA Modulizer. The VOILA Modulizer searches for a large number of relevant patterns, both common and complex.

Each set of events is delimited on the basis of AS "modules" found by MAJIQ in each analyzed gene. Modules refer to areas of the splice-graph between single entry (one junction path, diverges to two or more) and single exit (all junction paths converge back to one).

Inside each of the AS "modules" detected by the modulizer, smaller AS "events" (sub patterns matching specific known organizations of junctions or introns) are then categorized. Currently, the list of potential patterns we match to find an event is fixed to a specific set, which can be found in Supplementary Fig. 8a. All events which do not match any known splicing pattern are dumped to an "other" category which may be of possible interest in rare cases.

Modulizer supports any number or combination of MAJIQ experiments as input, in the form of PSI, dPSI, and/or HET VOILA files. These are used for narrowing modules to form around junctions/ introns we find relevant, as well as to verify which AS modules and AS events are changing or non-changing, based on coverage, Ψ, and differences in Ψ (ΔΨ). All filters may be disabled or adjusted.

At a high level, Modulizer uses a sequential pipeline for filtering and assembling output. First, all junctions and introns are read, and any which do not pass the reads, PSI, and/or dPSI thresholds are immediately removed from consideration. Then, using the remaining introns, and junctions, Modulizer identifies AS modules by looking for genomic locations with single entry/single exit as previously described. Then, Modulizer filters and removes modules which do not pass criteria such as not being sufficiently changing, lack of LSVs, or being constitutive. After filtering, Modulizer performs pattern matching for each AS event type on each AS module to identify all component AS events. Finally, Modulizer scans the input VOILA files for relevant quantifications in order to produce output TSV files for each individual AS event type, a high-level summary of all events found in each discovered module, and a summary of quantifications per module suitable for generating a heatmap according to the user's filtering criteria (e.g., the shortest discovered junction within the AS module to represent the inclusive AS product, the most changing junction in the AS module from HET and/or dPSI inputs, etc.).

## Sample selection from GTEx

We selected from GTEx in the following way. We required all samples to have a RIN score of greater than 6. For performance evaluation we chose to evaluate a comparison between cerebellum and skeletal muscle. We randomly selected 150 samples from both tissues, excluding the same donor from being selected in both tissues (Supplementary Data 2). For the brain subregions analysis, we selected all samples in GTEx v8 associated with brain tissue (not including pituitary gland). Samples were downloaded as FASTQ or as BAM and converted to FASTQ depending on when they were released. Samples that were part of v7 are available on SRA, so they were downloaded using SRA Tools (v2.9.6) as FASTQ files. New samples from the v8 release were only available as BAMs on the cloud, so they were downloaded using gsutil (v4.46) and converted to FASTQ using samtools (v1.9).

## Simulated RNA-seq as ground truth

We used the expression quantification data from the GTEx v8 release as the basis for our simulations. Briefly, we downloaded publicly available gene- and transcript-level quantification tables for GTEx v8 from the GTEx portal (https://www.gtexportal.org/home/datasets). To match how the GTEx consortium performed these analyses, we downloaded the GRCh38 build of the reference genome sequence and gene models from v26 of the GENCODE annotation.

We selected 300 samples from GTEx to serve as the basis for 300 simulated samples, each real sample providing the expression distribution underlying one simulated sample (Supplementary Data 3). To run BEERS, we first need to prepare four configuration files that are customized for the desired dataset: geneinfo, geneseq, intronseq, and feature quants. The geneinfo, geneseq, and intronseq files define the

structure and sequence information for each simulated transcript. As a result, these three files are determined solely by the choice of reference genome build and annotation. The feature quant files are specific to each individual sample and define a distribution of transcript-level expression. First, we used the genome sequence and gene models to create the geneinfo, geneseq, and intronseq files. Since the genome is fixed across all simulated samples. We used the same set of these files to simulate all GTEx-derived samples. Next, we extracted TPM values for each sample from the GTEx transcript quantification table and used these distributions of TPM values to generate separate BEERS feature quant config files for each simulated sample. Lastly, to determine the total number of reads to simulate for each sample, we used the gene-level quantification file to count the total number of gene-mapping reads in each GTEx sample.

To simulated strand-specific reads with uniform coverage across no errors, substitutions, or intron retention events, we ran the BEERS simulator using the following command-line options: `-strandspe-cific -outputfq -error 0 -subfreq 0 -indelfreq 0 -intronfreq 0 -palt 0 -fraglength 100,250,500`.

We transformed ground-truth transcript abundances into ground-truth splicing quantifications for each splicing quantification tool, taking into account the tools' differing definitions of splicing events. First, we defined ground-truth abundances for each exon or junction by adding the abundances of all transcripts including the exon or junction. Then, for each tool, we adopted their splicing event definitions, mapping the exon/junction abundances to compute their splicing quantifications.

**MAJIQ.** MAJIQ reports splicing quantifications with respect to LSVs. Therefore, ground-truth values for PSI were calculated by dividing the ground-truth abundance of each junction by the sum of the ground-truth abundances for all junctions in each LSV.

**rMATS.** rMATS reports a different format file per event type. But since all of them are classical binary event types, all can be reduced to two paths events, inclusion and exclusion. Each file contains the exon that defines each of the ways, so we calculate the $\Psi_{gt}$ as inclusion/(inclusion + exclusion) using the exon transcript combination to get the exons ground-truth abundances for all junctions in each LSV.

**LeafCutter.** LeafCutter reports splicing quantifications with respect to intron clusters composed of several junctions. Ground-truth values for LeafCutter's splicing ratios were calculated using ground-truth junction abundances, similar to MAJIQ.

**SUPPA2.** SUPPA2 reports classical events similarly to rMATS. So the approach we use here is similar to that tool. The main difference is that SUPPA2 reports the junctions coordinate in each one of the paths, so we use those junctions ground truth quantification to obtain the $\Psi_{gt}$ as inclusion / (inclusion + exclusion).

**Whippet.** Whippet outputs a psi.gz that contains the psi quantification of an event. That PSI is their formulation of the quantification from inclusion and exclusion paths. Unlike SUPPA2 or rMATS, Whippet combines a set of junctions to define a path, emulating in that way a transcript (or a portion of it). So, in order to find $\Psi_{gt}$ of those paths, we look for those transcripts that include all the junctions (and virtual junctions). We combine the expression of those transcripts to find the $\Psi_{gt}$ of each path.

## RNA-seq sample preprocessing before splicing analysis

We aligned RNA-seq reads from real and simulated GTEx samples to the human genome for splicing analysis with MAJIQ and other tools using the following procedure. Simulated GTEx samples were generated as pairs of FASTQ files. We performed quality and adapter

trimming on each sample using TrimGalore (v0.4.5). Some tools require reads aligned to the genome. For these tools, we used STAR (v2.5.3a) to perform a two-step gapped alignment of the trimmed reads to the GRCh38 primary assembly with annotations from Ensembl release 94. Other tools required transcript quantifications relative to annotated transcripts. For these tools, we used Salmon (v0.14.0) using the trimmed samples to estimate transcript abundances.

## Performance evaluations

We wrote a package of evaluation scripts, called validations-tools, in order to compare MAJIQ in terms of speed, memory footprint, accuracy, and reproducibility for each one of the following tools: rMATS, LeafCutter, SUPPA2, and Whippet. This package was written to allow future users to not only reproduce our results but to easily add future tools and repeat these kinds of analyses with different datasets.

We adjusted the tools parameters following recommendations by each tool's authors. Specific parameters are listed in Supplementary Data 4. For these comparisons, we evaluated the methods' computational efficiency and ability to identify splicing differences.

First, we evaluated computational efficiency of the different methods. We evaluated computational efficiency in terms of runtime and peak memory usage. Not all tools provide an extensive log of their execution, so, in order to measure wall time and memory usage, we used the output of `/usr/bin/time -v`. We ran each method for all pairs comparisons between 10 groups with increasing sample sizes on an Ubuntu Linux environment with 32 cores (Intel Xeon 2.7 GHz and 64 GB RAM).

Second, we evaluated the different methods' performance in quantifying splicing differences on simulated and real datasets. On the simulated datasets, where we know ground-truth differences in splicing between transcripts, we calculated true and false positive rates for the identification of splicing differences by each method. However, on real datasets, where no ground-truth is available, it is not possible to calculate true or false positive rates. Instead, we evaluated two metrics, reproducibility ratio (RR) and intra-to-inter ratio (IIR), on real (and simulated for comparison) data. The first metric, RR, measures the internal consistency of differential splicing tools. This internal consistency is reflected in the assumption that each tool should identify roughly the same events when repeating a comparison between two groups using different samples. We quantify this by performing two such comparisons and computing the fraction of the top $n$ differentially-spliced events in the first comparison that are also in the top $n$ events of the second comparison. This produces a "reproducibility-ratio" curve, RR($n$) for the method as a function of the number of top events. If the first comparison yields $N$ "significant" events, RR(N) is called the reproducibility ratio. For the specific case of MAJIQ, we note that in order to comparisons of LSV-type events more comparable to classic AS events such as used by rMATS, we filtered out overlapping LSVs (i.e., those that share junctions) in order to avoid double-counting classic AS events. For example, a classic exon-skipping event would have matching source and target LSVs that overlap. However, we note that this filtering only reduces $N_A$ but does not affect the reproducibility curves (apart from extending to a different value of $N_A$) (Supplementary Fig. 7). Although reproducibility of a method on real data is a scientifically important goal, it is not a sufficient goal because highly biased methods can be highly reproducible. To address this limitation, the second metric, IIR, is based on the principle that comparisons between (inter-) two groups should have many more significant events than comparisons within (intra-) a group. Furthermore, significant events within the group are likely false positives. This is quantified by computing the ratio of the number of significant events from an intra-group comparison to the number of significant events from an inter-group comparison. We evaluated these metrics for each tool with varying sample sizes to identify which methods outperformed each other in different settings.

**Event-level evaluations.** In these evaluations we check reproducibility and accuracy of reported differentially spliced events by the various tools shown in Fig. 2. As we describe in the main text, each tool defines alternative splicing events differently so that direct comparison of the events or their number between tools is not possible. Thus, when using real data each method was assessed by its own set of reported events to compute reproducibility ratios (RR) and intra-to-inter ratio (IIR) as in Fig. 2d, e.

In contrast, when using GTEx based simulated data we do have the "ground truth" (denoted "gt" below) for the abundance of each transcript. We thus use these values to summarize $\Psi$ and $\Delta\Psi$ observed in each method reported AS events and assess accuracy using the following definitions:

- True Positive: $\max \Delta\Psi_{tool} \geq 20\%$ and $pvalue_{tool} \leq 0.05$ and $\max \Delta\Psi_{gt} \geq 20\%$
- True Negative: $\max \Delta\Psi_{tool} < 5\%$ and $pvalue_{tool} > 0.05$ and $\max \Delta\Psi_{gt} < 5\%$
- False Positive: $\max \Delta\Psi_{tool} \geq 20\%$ and $pvalue_{tool} \leq 0.05$ and $\max \Delta\Psi_{gt} < 5\%$
- False Negative: $\max \Delta\Psi_{tool} < 5\%$ and $pvalue_{tool} > 0.05$ and $\max \Delta\Psi_{gt} \geq 20\%$
- Ambiguous: all other cases (when either $\Delta\Psi \in [5\%, 20\%]$ or when $\Delta\Psi$ and pvalue reported by the tool conflict),

where max is taken over all junctions/introns that belong to each AS event.

The above definitions were used to assess accuracy at the event level for each method, as shown in Supplementary Fig. 2, and also served as the base for gene level evaluations described below.

**Gene-level evaluations.** To facilitate more direct comparison between the different methods shown in Fig. 2 we aggregated each tool AS events and their respective annotation as TP, TN, FP, and FN as given above to assess gene level performance. Naturally, gene level labels of TP, TN, FP and FN are defined based on the events they contain. The gene-level labels are easy to define as positive or negative when all AS events embedded in it are considered positive or negative, respectively. The problem arises when a gene has some of its events as false positives and false negatives. In that case, we prioritize the labels according to the following order: FP, FN, TP, TN. This means for example that an occurrence of a false positive event in a gene (according to the method's specific event definition) would be counted as a false positive gene even if some other events were correctly labeled as true negative or even true positives. The rationale for this prioritization is that (a) positive events are expected to be rare and (b) we care the most about trying to validate or follow up on wrong hits (false positives) followed by missing true changes (false negatives).

## Testing different running configurations

The extensive evaluations described in this work require running different tools, each with its own settings. This requires addressing key pain points in the computational and genomics research, regarding both reproducibility and fair/accurate representation of competing software (see, for example, ref. [34]). With respect to the former, we produce the validations-tools package, supply all synthetic dataset used here and processed MAJIQ files (see "Data availability" and "Code availability" sections) so that our analysis can be both reproduced and extended to include other tools. With respect to fair comparisons, we believe that it is unrealistic for authors to try an extensive set of non-default parameters for all competing programs they are evaluating, yet it is important to make a concrete effort to have each method executed correctly. Specifically, we first strive to use up to date software. Second, if we run into specific problems/issues we are unclear about we contact the authors directly. During our work we ended up contacting the authors of Whippet, LeafCutter, and SUPPA and updated our evaluations several times as new versions/bug fixes were released. We

generally use software according to the authors specifications unless these specifications seemed problematic for a specific assessment task such that these would clearly degrade the methods' performance. In such situations, we try to assess sensible alternatives and report the best performance we find. Our efforts with respect to each methods configuration are reported below.

**LeafCutter.** For LeafCutter, default parameter were used with standard *p* value < 0.05. However, LeafCutter's default execution seems to have been motivated by sQTL analysis and did not include any explicit threshold on dPSI, only on the calculated *p* value. We noticed very high false positives (reflected through high IIR values) when using this approach. Furthermore, for tasks where the ground truth is set by defining a threshold on dPSI it seems inappropriate to use a *p* value only threshold by LeafCutter when its output also includes dPSI. We therefore deviated from the original authors' execution and report results for events that pass both *p* value and the dPSI threshold. When it comes to ordering events (as required for RR plots) we again found ourselves without author specified guidelines. We therefore tried both options that pass both filters: order by dPSI or order by *p* value (see Supplementary Fig. 6). We report results for the setting which gave better performance.

**rMATS.** For rMATS, default parameter were used with standard *p* value 0.05 and dPSI cut-off as described in relevant sections. For ranking events in RR plots, we applied both filters (*p* value and dPSI), tried ranking by either (see Supplementary Fig. 6), reporting the better results.

**SUPPA.** For SUPPA default parameter were used with standard *p* value 0.05 and dPSI cut-off as described in relevant sections. For ranking events, we applied both filters (*p* value and dPSI) and tried ranking by either (see Supplementary Fig. 6), reporting the better results. We also implemented a parallelization script to run the software on 16 threads (denoted as "SUPPA2(×16)" in Fig. 2a) so that it can compare more favorably to other methods that have multi-threading built-in. Running time out of the box without the parallelization script is reported Supplementary Fig. 1.

**Whippet.** For Whippet, we ended up needing to test several different configurations. First, after contacting the authors we found that the evaluations performed in Sterne-Weiler et al. applied the following filters: First, events were required to have a CI < 0.1 in ALL samples. The CI (confidence interval), serves as a measure of how sure Whippet is in the quantification. For small datasets, typically composed of biological replicates, these thresholds make sense as these should leave users with a confidently quantified set. However, for large heterogeneous data as assessed here, we found these thresholds to be overly restrictive as many AS events may end up not passing the CI < 0.1 in at least some of the (many) samples. Left as is, these thresholds result in poor performance. We therefore left Whippet confidence parameter per event in a sample as is (CI < 0.1) but required only 50% of the samples to pass that filter - analogous in spirit to MAJIQ's default filter of requiring an event to be quantifiable in at least 50% of the samples in the group. We also tried an 80% threshold but 50% seemed better.

An issue that came up during the review of the manuscript is how Whippet's time and memory performance should be presented compared to other methods. Here, we preferred to use a multi-core machine and report wall-time as we view this is a more realistic representation of performance than CPU-time. The latter, commonly used for comparing algorithms efficiency is arguably a more theoretical measure as it does not capture I/O bottlenecks or improved parallelization implementation of a specific method. However, since Whippet does not support multi-core usage out of the box this assessment approach made it appear slow even though the algorithm was efficient. To resolve this issue we implemented a simple script that distributes Whippet jobs across multiple cores. We tested performance using 4, 8, and 16 cores and found 16

cores performed slightly better than 8 cores, likely due to I/O bottleneck. We reported performance using this script in the Fig. 2a (denoted "Whippet(x16)") and running time without the parallelization script in Supplementary Fig. 1.

Another consideration in Whippet's testing regards the complexity of the events reported by Whippet. Whippet uses the parameter *K* to denote log2 of the number of possible paths through an AS event. A classical event (e.g., cassette exon) would have $K = 1$, while more complicated events would have $K \geq 2$. In Sterne-Weiler et al., the authors evaluated Whippet (and all other competing methods) only on $K < 2$ events, i.e., only on simple "binary" events. This decision stemmed from the desire to create a common base with methods such as rMATS which only operate over classical (binary) AS events. Here, we tested Whippet with both complex and binary (i.e., no $K < 2$ filter) and reported evaluations do not include the original $K < 2$ filter. Our rationale for this was that many methods (Whippet, LeafCutter, MAJIQ) can now handle complex events. Furthermore, it is important to include those since in many cases the ability to detect and report such events is part of the highlights/selling points of such methods and users should be able to assess performance in the context of what the method actually gives them (i.e., both complex and binary events). As can be expected, Whippet's metrics with no filtering of $K \geq 2$ improved significantly in terms of the total number of events reported in Fig. 2b (see also Supplementary Data 5–8).

Yet another consideration when evaluating methods that detect very different event types is which types of AS events to include/exclude. For example, Whippet detects events of type transcriptional start (TS), transcriptional end (TE), and back-splicing (BS), which other methods do not. Similarly, MAJIQ is the only method that detects unannotated intron retention. After contemplating much about how such differences should be considered we converged to the following procedure: In tests where we use the methods "native language" of AS events it reports we kept all event types (but see restriction applied against MAJIQ to avoid "double counting" and inflating its stats below). This applies to RR and IIR where the logic is to assess each method by its own set of events in terms of what the user will get. However, when trying to create a more uniform assessment at the gene level rather than at the event level, as we did using synthetic data, then we excluded unique event types (i.e., Whippet's TS/TE/BS and IR). We did not exclude complex events here as most methods detect those (see above) and the criteria applied is a change by 10% or 20% based on the methods own event "language" (binary/complex etc.)

**MAJIQ.** For MAJIQ, we did not perform any special tweaks/parameter optimization, running it with default parameters. The only addition is that we used the flag "show-all" in order to actually get in the output files all events assessed by MAJIQ rather than only events that pass the default (conservative) thresholds of dPSI > 20% with a posterior probability > 0.95. Failing to using this flag while trying to assess MAJIQ's performance over all events across different thresholds (e.g., ROC plots) has previously led to severe misrepresentation of MAJIQ's performance[5,34].

The only tweak we performed for MAJIQ's evaluation is to exclude overlapping LSVs when assessing performance at the event level (i.e., for RR and IIR). This is because classical events reported by other software (e.g., rMATS) can be captured by overlapping LSVs such that counting/reporting those would inflate MAJIQ's stats unfairly. For example, a highly significant change in a cassette exon detected by rMATS would be captured by two LSVs in MAJIQ: a single source LSV and a single target LSV. Thus, when reporting RR and IIR we rank all LSV as usual but an LSV is disregarded if an overlapping LSV has already been included in the RR curve. The effect of discarding overlapping LSVs on MAJIQ's performance can be seen in Supplementary Fig. 7.

To summarize, we did not try to exhaustively test all parameter combinations for any specific software (MAJIQ included). When using default parameters did not make sense or did not match the test's

assumptions we tried to give each method a fair representation, exploring additional parameters/filter settings to the best of our understanding, and reporting the best results. Finally, we note that all our testing data, test scripts and parameters are documented and included as a package for future use.

### GTEx brain subregion analysis

**MAJIQ HET and VOILA Modulizer on GTEx brain subregions.** MAJIQ HET was run on all 78 unique pairwise comparisons of GTEx v8's 13 brain tissue groups, and the results were visualized with VOILA. Significant LSVs were those considered to be those containing at least one junction or intron with an absolute difference in group median $\mathbb{E}[\Psi]$ values of 20% or more between the two tissue groups and all four HET statistics (Mann–Whitney, InfoScore, TNOM, and $t$ test) with $p < 0.05$.

The VOILA Modulizer was run on the resulting outputs with the following options: `–decomplexify-psi-threshold 0.05` to remove all junctions and introns from the splicegraph that had tissue group median E(PSI) of less than 5% across samples for every group; `–show-all` to include all AS modules and AS events in the output, not just those meeting the changing criteria. Default values were used for other options that flag changing AS modules and AS events in the output. For changing: a minimum absolute median difference in PSI between groups of 20% or more for the primary threshold and a $p$ value of less than 0.05 across all four MAJIQ HET statistics (Mann–Whitney, Info-Score, TNOM, and $t$ test). For non-changing: a maximum absolute median difference in PSI of 5% or less between groups; a maximum interquartile range in PSI within a group of 10% or less; and a $p$ value of 0.05 or greater across the MAJIQ HET statistics.

**PSI-based AS module and AS event counts across the brain.** Counting of AS modules based on the initial PSI simplification across the 13 brain tissue groups was done by parsing the resultant VOILA Modulizer summary file. This file is organized by AS module and lists the number of each of the 14 AS event types, outlined in Supplementary Fig. 8a, contained in each. AS modules were classified and counted based on the presence or absence of each of the 14 AS event types. Certain AS event type definitions overlap. Specifically, every tandem cassette exon containing AS module will also contain a multi exon skipping AS event and every putative 5' or putative 3'ss AS module will also contain an intron retention event. In these cases, the additional, partially redundant AS event type was added to the AS module classification if and only if their count within the module was larger than the count of the AS event they overlap with. For example, for an AS module to be classified as containing both tandem cassette exon (TCE) and multi exon skipping events (MES), the number of MES events within the module must be greater than the number of TCE events.

The pan-GTEx analysis of intron retention, AS modules, and AS events (Supplementary Fig. 9) was carried out in the same way, except using a subset of up to 15 high quality samples to represent each tissue that we defined previously[42], which came from unique donors with high RNA integrity numbers. The Modulizer was run on either all GTEx tissues together or a subset of individual tissues, one at a time, to assess how AS module and AS event definitions differ across different tissues with varying degrees of IR.

**Cerebellar AS module and AS event definitions.** Given the large number of LSV-based splicing differences between the two GTEx cerebellar tissues (cerebellum and cerebellar hemisphere) and the other brain subregions according to MAJIQ HET comparisons (Supplementary Fig. 10a), we wished to define AS modules and AS events based on these comparisons. These two cerebellar tissues were derived from sampling in duplicate. Cerebellum (and Cortex) were sampled at the time of non-brain organ sampling and preserved in PAXgene tissue fixative solution while Cerebellar Hemisphere (and Frontal Cortex) were sampled later from the remaining frozen whole brain, along with

other brain subregions, at the brain bank[43]. Therefore, we focused our analysis on AS modules and AS events that displayed changes between both cerebellar tissues and one of the other subregions. For example, a cassette exon AS event would have to be labeled as changing according to the VOILA Modulizer filters (minimum absolute median difference in PSI between groups of 20% or more for the primary threshold and a $p$ value of <0.05) in both cerebellum versus cortex and cerebellar hemisphere versus cortex to be counted. We defined all such consistent, changing cerebellar AS events from the 14 AS event files output by the VOILA Modulizer and used these to count the number of modules containing each AS event type or combination of types.

**Cerebellar cassette exon regulatory analysis.** To perform regulatory analysis around exons with differential cerebellar inclusion patterns we first defined a high confidence set of cassette exons (CEs) by applying additional filters to those described above. In addition to the primary filter of an absolute median difference in PSI of 20% or more between a cerebellar tissue and another brain subregion for one junction in the CE event, a secondary threshold of an absolute median difference in PSI of 10% or more was enforced for all four junction quantifications of the CE (i.e., the inclusion source LSV junction quantification, the inclusion target LSV junction quantification, and the shared exclusion junction quantified in both the source and target LSV). Next we enforced that the direction of change between the two exclusion junction quantifications and the two inclusion junction quantifications agreed in their direction of change in cerebellar versus other tissues. If both inclusion junction qualifications increased in cerebellar tissues and both exclusion junction quantification decreased, this was considered a cerebellar inclusion CE events. The opposite directions were considered cerebellar exclusion CE events. Non-changing CE events were defined as those flagged as non-changing by the VOILA Modulizer in every comparison of both cerebellar tissues versus the other 12 brain tissues. For CE subset analysis, CE with intron retention (IR) events were those where one or more of the CE junctions was also involved in a changing IR event in cerebellar versus other tissues. CE with no IR events were those CE events that came from modules without any IR events detected.

For sequence analysis we extracted GRCh38 sequences for intronic regions 300 nucleotides (nts) upstream and 300 nts downstream of every CE in each set. We calculated $Z$-scores by comparing the occurrence of each hexamer in the upstream intronic region in each cerebellar set of regulated CEs versus the non-changing set of CEs. This was repeated for the downstream intronic region as well. $Z$-score analysis was carried out in the same way for upstream and downstream distal intronic regions that were >500 nts from the splice sites of both the alternative exon and the flanking constitutive exons.

Motif maps were generated to visualize position specific enrichment of particular hexamers of interest. Each hexamer, or set of hexamers, were searched for over sliding windows of 20 nts in the splice site proximal regions around the CE (i.e., intronic region 300 nt upstream of the 3'ss plus 50 nt downstream and 50 nt upstream of the 5'ss plus the intronic region 300 nt downstream). The frequency of occurrence was determined in each CE set and plotted using a running mean of 5 nts for smoothing.

RNAmaps for CLIP based binding of QKI were plotted in a similar way over the same splice site proximal regions. BED narrowPeaks files were downloaded for ENCODE eCLIP data[44] from encodeportal.org for QKI in K562 cells (accession ENCSR366YOG) or QKI in HepG2 cells (accession ENCSR570WLM) and replicate files were concatenated. BED narrowPeaks for uvCLAP data for QKI-5 in HEK293 cells[45] were downloaded from GEO (accession GSE85155) and lifted over from GRCh37 to GRCh38. These peak coordinates were overlapped with CE splice site proximal regions and the frequency of occurrence was assessed over the various cerebellar CE event sets at each position proximal to CE splice sites.

To examine if cerebellar CE splicing patterns showed evidence of QKI regulation, we utilized QKI shRNA knockdown RNA-seq data from HepG2 cells (accession ENCSR330YOU) generated by ENCODE[44]. We processed these samples as described previously[2]. Briefly, batch correction was performed using Moccasin and splicing changes upon QKI knockdown were quantified by comparing knockdown samples to all HepG2 control samples using MAJIQ HET. Significant changes upon knockdown were those with an absolute median difference in expected PSI of ≥20% and a Mann–Whitney two-sided $p < 0.05$. We intersected QKI regulated splicing changes with cerebellar CEs (defined above) and calculated the fold enrichment and significance of overlap of QKI regulated splicing with cerebellar regulated versus nonregulated CEs using a two-tailed binomial test. We further stratified these overlaps by the direction of splicing change in the cerebellar tissues versus QKI knockdown to define overlaps that were consistent with our model that decreased QKI expression promotes cerebellar cassette exon patterns. Therefore, overlaps that were consistent with our model were the cerebellar exclusion CEs with decreased CE inclusion upon QKI knockdown or cerebellar inclusion CEs with increased CE inclusion upon QKI knockdown. Those that were inconsistent with our model were cerebellar exclusion CEs with increased CE inclusion upon QKI knockdown or cerebellar inclusion CEs with decreased CE inclusion upon QKI knockdown.

### Reporting summary

Further information on research design is available in the Nature Portfolio Reporting Summary linked to this article.

## Data availability

The raw simulated RNA-seq data for cerebellum and skeletal muscle generated in this study have been deposited in the GEO database under accession code GSE222044 at https://www.ncbi.nlm.nih.gov/geo/query/acc.cgi?acc=GSE222044. Raw GTEx data used for the analyses in this manuscript are available in dbGaP under accession code phs000424.v8.p2 at https://www.ncbi.nlm.nih.gov/projects/gap/cgi-bin/study.cgi?study_id=phs000424.v8.p2. ENCODE raw RNA-seq for QKI knockdown and eCLIP peaks were downloaded from https://www.encodeproject.org/ under accession codes ENCSR366YOG, ENCSR570WLM, and ENCSR330YOU. Processed data and code to reproduce figures have been deposited in a Zenodo repository available at https://doi.org/10.5281/zenodo.7508313.

## Code availability

The code for MAJIQ and VOILA are available for academic/non-commercial use at majiq.biociphers.org. Licensing information for commercial use can be found at majiq.biociphers.org/commercial.php. The code for validation tools is available at bitbucket.org/biociphers/validations_tools.

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

## Acknowledgements
We thank Dr. Elizabeth J. Bhoj for her input on the final manuscript and co-advising JKA during the project. We also thank Mr. Christopher J. Green for his contributions to an earlier version of VOILA.

## Author contributions
Y.B. conceived the project. J.V.-G., J.K.A., S.S.N., and Y.B. developed and tested the methodology for MAJIQ HET. J.K.A. improved the quantification, sampling and statistical testing in the final MAJIQ HET implementation. J.V.-G. did the initial work porting Python code from MAJIQ v1 to MAJIQ v2. J.V.-G. and Y.B. developed the new IR quantification algorithm. J.K.A. formalized the approach for bootstrapping readrates. J.V.-G. and J.K.A. implemented the updated MAJIQ Builder and MAJIQ PSI. J.V.-G. implemented MAJIQ dPSI. S.J. implemented the VOILA viewer with input from M.R.G., C.M.R., and A.J. S.J., C.M.R., M.R.G., A.J., and Y.B. developed the methodology for VOILA Modulizer. S.J. implemented the VOILA Modulizer. N.F.L. and G.R.G. generated the simulated RNA-seq data for the performance comparisons. J.V.-G. carried out the performance comparisons vs other tools. M.R.G. performed the modules analysis with input from C.M.R. M.R.G. conceived and carried out the brain subregions analysis with input from Y.B. Y.B., M.R.G., and J.K.A. wrote the final manuscript with input from S.J. and J.V.-G. C.M.R. and A.J. contributed equally to this project. All authors read and approved the final manuscript.

## Funding
National Institutes of Health grants R01 AG046544, R01 LM013437, R01 GM128096 (Y.B.), National Institutes of Health grant F30 HD098803 (J.K.A.), and Blavatnik Family Fellowship in Biomedical Research and National Institutes of Health grant F31 HL162546 (M.R.G.).

## Competing interests
The MAJIQ software used in this study is available for licensing for free for academics or for a fee for commercial usage. Some of the licensing revenue by the University of Pennsylvania goes to members of the Barash lab including Y.B., J.V.-G., J.K.A., S.J., M.R.G., and C.M.R. Otherwise, all authors declare they have no competing interests.
