## [Peer Review File · Nature Communications]

RNA splicing analysis using heterogeneous and large RNA-seq datasetsREVIEWER COMMENTS

Reviewer #1 (Remarks to the Author):

Here, Vaquero-Garcia et al present MAJIQ v2, a suite of algorithms for detection, quantification and visualization of alternative splicing. In particular, the package is designed to be able to handle thousands of datasets, something that is necessary given the many large collections available today. The authors present careful benchmarking against existing methods which strongly supports the advances of their new algorithm. Overall, I find the paper well-written and it is easy to follow. Nevertheless, I would like for the authors to address the following major concerns:

For Fig 2d, I assume that the samples were selected randomly (e.g. 3 liver samples and 3 cerebellar) from the GTEx? If so, could the authors do this multiple times to calculate confidence intervals for each method? The same goes for the IIR in 2e.

The evaluation using RT-PCR is a strong aspect of the manuscript. However, I would like the authors to carry out a statistical test (via Fisher transformation) to compare the correlation coefficients between LeafCutter and MAJIQ to determine if the performance difference is statistically significant.

To allow users to more easily get started with MAJIQ it would be very helpful to have a very simple example/vignette where the required files can be downloaded from your website. Having an example (along with files) that is already working is in my experience very helpful when one has to get a new software to work with one's own data. Here, I was able to install the software without any problems and I started to follow the quick-start guide. However, I did not get very far as I could not figure out how to set up a configuration file for the Builder. The reason was that the link https://biociphers.bitbucket.io/majiq/quick.html#conf_file does not seem to be pointing to the correct place and I could not find the instructions for how to set up this file by scrolling up and down the manual pages.

On p15 the authors discuss high levels of intron retention in neuronal tissues and they conclude that this is consistent with previous finding. However, as they have GTEx data available which is presumably of better quality than what was used in ref 19, it would be useful if they carried out this comparison on other tissues from GTEx to get a better and more relevant benchmark. Comparison against other tissues would also be helpful to better understand the results reported in the next paragraph. I would like to know if these uncommon AS modules are enriched in the brain or if these types of events are common across all tissues. As it stands it is hard to evaluate the significance of this result as there are no points of comparison.

The authors report downstream enrichment of the RBFOX motif. However, as it has been shown by Lovci et al (<https://www.nature.com/articles/nsmb.2699>) that distal motifs (>500 nts) are common for these motifs, I think that the authors need to comment on this result as the implication is that their search can only uncover half of the relevant motifs.

The results presented regarding QKI in Fig 4 are quite interesting, but they are all correlative. However since this paper is about a computational method, I am not requesting that they perform their own experiments, but they should be able to validate some of these predictions using ENCODE data from shRNA experiments followed by RNA-seq (e.g. <https://www.encodeproject.org/experiments/ENCSR256PLH/>).

Minor concerns:

The low run time of MAJIQ is obviously a nice feature. However, it would be helpful to at least have rough breakdown between the time for MAJIQ Build and Quant to get a better idea of how much time is saved by the incremental build strategy.

It is always hard to summarize and represent the performance of different methods. However, I think it would be a good idea to include either F1 or MCC in Fig 2b to also have a measure which combines more aspects of the confusion matrix than FDR and FNR does.

It would be helpful to have labels on the axes in 2d

Some of the text in 3a, b is really small. I had to zoom in on the pdf and this should not be a requirement to be able to read the paper.

On p17, line 4-5 there are two downstream but one of them should probably be changed to upstream.

It is not for me to decide on these matters, but in my opinion I find the discussion to be on the long side and it is my recommendation to shorten it.

On p29 it is not clear how the three priors for dPSI are weighted. Are they all given the same importance?

Reviewer #2 (Remarks to the Author):

Review of NCOMMS-22-07149

In the manuscript, entitled "RNA splicing analysis using heterogeneous and large RNA-seq datasets", Vaquero-Garcia et al. present an update to their previously published splicing software, MAJIQ, showcasing new tools applied to analyze splicing across large cohorts of RNA-seq samples. These, among other things, are designed to elegantly handle the problems associated with thousands of unannotated minor splicing events, which is a challenge in the field. They compare their software to existing methods, and show that there is significant benefit to using their method which is tailored specifically to the task of comparing across many diverse samples. While the specific statistical tests they benchmark are

previously described, and the MAJIQ v2 software is built upon their prior publication, I think the suite of tools they present here has sufficiently diverged from the original publication enough to warrant interest from a broad readership as to its new capabilities and applications.

Overall, I also feel that the analytical and benchmarking contributions presented here are valuable and will be of interest and an asset to the field. That said, there are several analyses related to Fig. 2 that could be strengthened and/or require changes in order to support the author's claims (e.g. see #3, #7 and #10 below), and additional points for discussion that should be clarified and included or addressed prior to publication (e.g. see #8 & #9).

Please find my list of specific comments below:

(1) I applaud the authors for formalizing their benchmarking analysis as a package for the community. I think it speaks to their efforts to make fair comparisons, enable reproducible science, and move the field forwards in a positive direction. However, at the time of this review the link to ``bitbucket.org/biociphers/validations-tools`` was not public. It would be helpful if this code could be made available to the reviewers, so that any technical details within it absent from the methods text can be checked.

(2) For example, in Fig. 2 some of the evaluated programs quantify tandem alternative poly-adenylation sites or tandem alternative promoters, while others do not. (A) Are these event types being simulated in the synthetic data experiments? (B) If not, are these events filtered prior to evaluation to ensure a more stable and fair comparison? The authors should include these details in the methods.

(3) In the author's comparison of runtime/memory between software packages (Fig. 2a), they compare programs that start from raw-sequencing reads in ``fastq`` format alongside those that start from pre-aligned reads in ``bam`` format, without including the alignment time for producing the bam-file in this comparison. For example, Whippet's algorithm is based on internal alignment of junction flanking k-mers from raw-reads directly without external alignment software, while MAJIQ, LeafCutter, and rMATs all require pre-alignment-- a step the authors perform with STAR. Since both the runtime and memory requirements of this alignment step are frequently the most significant computational resources required for the entire start-to-finish RNAseq analysis process-- this discrepancy is a very relevant one. For example, in the text Whippet and SUPPA are presented as the two slowest algorithms (at 50-55 hours vs. 6 hours for other programs), whereas when starting from the same point of the RNAseq analysis in a fair head-to-head evaluation, Whippet and SUPPA are actually much faster than the methods requiring pre-alignment with STAR (assuming SUPPA is leveraging Salmon or Kallisto's speed; see 10.1016/j.molcel.2018.08.018).

Furthermore, it is standard in the field to include this alignment time in such comparisons, as can be seen in previous literature. For example, in the landmark papers for Sailfish: 10.1038/nbt.2862, and Kallisto: 10.1038/nbt.3519, external alignment-time is included in the analogous benchmarks of gene expression quantification software starting from fastq-files (Sailfish/Kallisto/Salmon) when compared to

programs that only perform the last RNAseq quantification step starting from pre-aligned bam-files (RSEM, Cufflinks, eXpress etc.). This is also how 4 of the 5 splicing programs evaluated in the current manuscript (eg. MAJIQ, Whippet, SUPPA, rMATs) have been compared in the literature previously (see above).

For the sake of transparency, consistency, and correctness, the authors should: (A) make clear the discrepancy in runtime/memory required for fastq-file vs. bam-file starting-points among programs in both the text and the figures, and (B) include the missing alignment runtime/memory of a standard aligner like STAR for a fastq-file starting point in the final comparison.

(4) In Table S4, the authors appear to utilize multicore processing with `number_of_cores=16` for [MAJIQ, rMATs, and LeafCutter], while [SUPPA and Whippet] do not support this. The authors should clarify whether the runtime benchmarks in Fig. 2a are "wall" time (ie. straight start-to-finish clock time ignoring the difference in # cores used) or "cpu" time (ie. additive cpu-time to account for differences using multiple cores). Given the scope of the manuscript towards analyzing large cohorts of heterogeneous samples (which could be run concurrently across multiple cores using single-core programs), multi-core capable programs should not be given any advantage in this manuscript, and "cpu" time should be used.

(5) It is unclear what the author's intention is with the Pg. 7 statement, "0.5-4Gb of memory, an amount readily available on a laptop", or how it fits into the manuscript. Following comment #3 above, the main computational step of start-to-finish RNAseq analysis that might not be compatible with a laptop, would be the alignment step, which the authors excluded (eg. STAR has considerable runtime and requires ~ 30 Gb of RAM, which most laptops cannot accommodate). Furthermore, the benchmarks in the manuscript were performed on a compute node with 32 cores (Intel Xeon 2.7GHz and 64GB RAM) rather than a laptop, obfuscating the statement.

(6) In Fig. 2b, the results look to be very consistent for each program across the six sets of groups: 5v5 through 50v50. If these are essentially redundant data, it might be worth plotting only a subset of these (like 5v5 and 50v50)?

(7) In Fig. 2b, I think FDR & FNR used here are important metrics to evaluate sensitivity and specificity (inversely). However, one expectation is that differences in evaluated performance might exist as a function of the specific threshold of significance used for each program. Have the authors considered plotting these metrics as a function of a sliding cutoff from a rank-based metric, such as deltaPSI? For example, this additional analysis would importantly reveal the effect of altering significance thresholds towards the outcome, and might more closely resemble standard practices from other areas of classification method evaluation (eg. ROC curves)? This would strengthen the author's claims that MAJIQ's methodology, rather than a specific threshold choice, is responsible for its elevated performance.

(8) For Fig. 2b-e & also in the discussion, the authors acknowledge the effect of significance threshold choice in the analysis of LeafCutter, claiming deltaPSI $>20\%$ improves its performance in their

evaluations of RR. However, it is not fully clear (A) whether they applied non-default parameter adjustments/filters to any of the other algorithms (like rMATS, SUPPA or Whippet), and if so, (B) whether or not these changes affected the performance compared to default? (C) Similarly, if all programs had the same deltaPSI >20% and probability/statistical thresholds-- were these always consistent with the author's documented guidelines/suggestions, or did they diverge for some programs? For example, a deltaPSI cutoff of >20% may improve LeafCutter and MAJIQ, but (D) could it decrease the performance of another program whose statistical test is already inherently more stringent?

While I do not believe it is the benchmarking author's responsibility to try an extensive set of non-default parameters for the competing programs they are evaluating, it would seem to be an important discussion/analysis point for the text-- particularly if some programs (like LeafCutter) are investigated both within and outside of defaults, but others are not (rMATS, SUPPA, and Whippet), or if some programs benefit in performance from non-default thresholds used, but others regress?

As a post-script to this comment -- I'd like to point out that Whippet-documentation for ``delta.jl`` provides no more than a loose guideline on significance criteria (probability > 0.9 and absolute deltaPSI > 0.1), with a note that while the quantification of PSI with ``quant.jl`` has been extensively benchmarked for accuracy, for differential splicing "no specific significance thresholds have been systematically optimized". Given the evaluations presented in Fig. 2b-e, it would seem that this lack of calibration negatively affects the out-of-the-box performance of `'delta.jl'`, and I commend the authors of Vaquero-Garcia et al. for their analyses, which point out this limitation.

(9) In the manuscript, previously described statistical methods are applied to MAJIQ in order to identify significant differences in MAJIQ's underlying PSI values across large heterogeneous sample sets (TNOM, InfoScore and Mann-Whitney U). Since some of these statistical methods are currently available with external implementations (eg. Mann-Whitney-U is a base function in most languages, and TNOM looks to be implemented in the ClassComparison R package), they could also be easily applied to any of the other methods compared in this manuscript (eg. LeafCutter, SUPPA, rMATS, Whippet). For example, in Fig. 4 the authors define significant events using a simple Wilcox test (which is standardly applied across research in many fields, including splicing).

While that choice of statistical test may not be unique to MAJIQ here, it is likely that the performance of any statistical test in question will be limited by the accuracy of the underlying single-sample splicing quantifications used (eg. PSI values). This makes the accuracy of single-sample splicing quantifications an important metric in evaluating the advances MAJIQ provides in relation to other software methods of comparable scope. The authors should consider including this as a discussion point.

(10) In Supplemental Fig. S2, (and the text on Pg. 10), the authors do compare the accuracy of single-sample PSI point-estimates to experimental values from RT-PCR for MAJIQ and LeafCutter, but do not include plots of these accuracy values side-by-side to the other programs benchmarked in Fig. 2 (eg. SUPPA, rMATS, Whippet). While the authors refer to their previous publication where they plot these

values for SUPPA and rMATs, Whippet is not included. (A) The authors could plot a summary of this data and discuss whether or not accuracy at this level (or lack thereof) is concordant with their results when benchmarking deltaPSI in Fig. 2.

Similarly, given the importance of single-sample splicing quantification accuracy (see #9 above), (B) the authors should assess the accuracy of single-sample PSI using the synthetic data they simulated for Fig 2 -- plotting PSI point-estimate vs. ground-truth PSI (summary panels of these analyses could perhaps join the others in Fig 2).

(11) Fig. 3 -- I think the Voila Modulizer and visualization tools look great. In general, visualization of splicing for large heterogenous cohorts is a weak point in the field, and a tool like this has a lot of value and should accelerate research. Towards this end-- the discussion mentions that a lab or consortia center could process data like GTEx once, and then add their own samples to it from future experiments. Since the one-time analysis of the entire GTEx cohort is a considerable endeavor in and of itself, another point to highlight here might be how easy it is for a single user to compare a few internally produced samples of interest side-by-side to all of the pre-analyzed GTEx data (either by uploading their data to a remote server that contains the data, or by downloading the pre-analyzed data if it is made available)?

Tim Sterne-Weiler

We would like to thank the reviewers for their detailed comments and suggestions. It was clear reading the comments that both reviewers took the time to carefully read and consider our manuscript and we were happy to find they both viewed it positively while giving ample feedback to improve the manuscript.

We apologize for the delay in getting the revision ready. We ended up requesting additional time from the editor as we had to work through two of the first authors graduating/leaving and a paternity leave. Nonetheless, we have worked hard to address the reviewers comments and believe the manuscript has been substantially improved thanks to their feedback. We hope the reviewers will find the revised manuscript suitable for publication and detail our response/edits/additions below.

REVIEWER COMMENTS

Reviewer #1 (Remarks to the Author):

Here, Vaquero-Garcia et al present MAJIQ v2, a suite of algorithms for detection, quantification and visualization of alternative splicing. In particular, the package is designed to be able to handle thousands of datasets, something that is necessary given the many large collections available today. The authors present careful benchmarking against existing methods which strongly supports the advances of their new algorithm. Overall, I find the paper well-written and it is easy to follow. Nevertheless, I would like for the authors to address the following major concerns:

For Fig 2d, I assume that the samples were selected randomly (e.g. 3 liver samples and 3 cerebellar) from the GTEx? If so, could the authors do this multiple times to calculate confidence intervals for each method? The same goes for the IIR in 2e.

Yes, samples were selected by random. We repeated this analysis multiple times to demonstrate the difference between methods are significantly different and results are robust to the selection of specific subsets (new Supplementary Fig. 3). We also point this out in the main text. We did not perform comprehensive confidence intervals analysis given how stable the results appear to subset selection, the distinct difference between the methods, and the formidable computational overhead. Previous work by us where we did CI assessment involving smaller data is inline with this as well (see Norton et al 2017 Fig5).

The evaluation using RT-PCR is a strong aspect of the manuscript. However, I would like the authors to carry out a statistical test (via Fisher transformation) to compare the correlation coefficients between LeafCutter and MAJIQ to determine if the performance difference is statistically significant.

Excellent point. We now include the RT-PCR stats in revised Fig 2f, scatter plots for all methods against RT-PCR (Supplementary Fig. 5) and have added the computation of statistical

significance for the differences. Specifically, we used Dunn and Clark's z procedure as implemented in cocor package to avoid issues of predictor intercorrelation.

To allow users to more easily get started with MAJIQ it would be very helpful to have a very simple example/vignette where the required files can be downloaded from your website. Having an example (along with files) that is already working is in my experience very helpful when one has to get a new software to work with one's own data. Here, I was able to install the software without any problems and I started to follow the quick-start guide. However, I did not get very far as I could not figure out how to set up a configuration file for the Builder. The reason was that the link https://biociphers.bitbucket.io/majiq/quick.html#conf_file does not seem to be pointing to the correct place and I could not find the instructions for how to set up this file by scrolling up and down the manual pages.

We apologize for these issues when trying to use the software. While submitting the manuscript we were in the process of overhauling the documentation. The updated documentation is available at <https://biociphers.bitbucket.io/majiq-docs/>. To handle the specific problem of knowing how to create the configuration file we have also added a dedicated page as part of the quick start guide to better explain how to create a configuration file (<https://biociphers.bitbucket.io/majiq-docs/getting-started-guide/builder.html>) and have also included links to our "command line builder" interactive web form (<https://biociphers.bitbucket.io/majiq-docs/commandbuilder.html>). This interactive web form asks users questions to automatically create a configuration file and command line arguments to suit their specific data / analysis interests. We agree that vignettes are incredibly helpful, so we have added a series of vignettes for each of the MAJIQ quantification modes (i.e. psi, dps, and het, see <https://biociphers.bitbucket.io/majiq-docs/gallery.html>) to the updated documentation.

On p15 the authors discuss high levels of intron retention in neuronal tissues and they conclude that this is consistent with previous finding. However, as they have GTEx data available which is presumably of better quality than what was used in ref 19, it would be useful if they carried out this comparison on other tissues from GTEx to get a better and more relevant benchmark. Comparison against other tissues would also be helpful to better understand the results reported in the next paragraph. I would like to know if these uncommon AS modules are enriched in the brain or if these types of events are common across all tissues. As it stands it is hard to evaluate the significance of this result as there are no points of comparison.

We agree with the reviewer that information about non-classical AS modules and AS event types across additional tissues beyond the brain subregions analyzed are useful and informative. To address these suggestions, we added an entire new supplementary figure (now Supplementary Fig. 9) with matching additions to the Results, Discussion, and Methods sections. Here we first analyze the amount of intron retention across all GTEx tissues. We find that cerebellar tissues are among the top tissues in terms of levels of intron retention (i.e. higher median percent intron retained, PIR, for quantified introns compared to other tissues (Supplementary Fig. 9a) and more quantified introns above a threshold value of 10% retention (Supplementary Fig. 9b)). Surprisingly, the other GTEx neuronal tissues are towards the bottom

of the tissue distribution. We have added text in the results section and discussion to comment and better frame these results in light of the literature, specifically Braunschweig et al. 2014.

Next we used the VOILA Modulzier on a subset of tissues representing different degrees of intron retention and find that both non-classical AS event types (Supplementary Fig. 9c) and complex AS module types (Supplementary Fig. 9d,e) we described in the brain-subregion analysis are similarly detected in these diverse tissues. Interestingly, the relative fraction of these AS events and AS modules seem similar to our brain region analysis, even when only considering a single tissue at one time with the Modulzier analysis. We conclude that these AS events and complex AS modules are relatively common across all tissues we analyzed.

The authors report downstream enrichment of the RBFOX motif. However, as it has been shown by Lovci et al (<https://www.nature.com/articles/nsmb.2699>) that distal motifs (>500 nts) are common for these motifs, I think that the authors need to comment on this result as the implication is that their search can only uncover half of the relevant motifs.

Yes, this is another good point. To address this concern we performed an additional Z-score analysis that focused on only distal intronic regions that were >500 nt away from any splice site involved in the cassette exon definition. Like the proximal intronic enrichment of UGCAUG downstream of cerebellar included cassette exons, we also find this motif enriched in the distal intronic regions of the downstream intron, but to a lesser degree than proximal intronic regions. Interestingly, we also find CA-repeat motifs to be the most enriched motif around cerebellar included exons in distal intronic regions both upstream and downstream. These motifs are putative binding sites for hnRNP LL, a protein that was identified as tissue-enriched in the cerebellum in an analysis of GTEx proteomics samples (Jiang et al., 2020). We added a new panel to the supplementary figure dealing with motif enrichments (Supplementary Fig. 11g in the revised manuscript) to summarize these results and discuss them in the results section.

The results presented regarding QKI in Fig 4 are quite interesting, but they are all correlative. However since this paper is about a computational method, I am not requesting that they perform their own experiments, but they should be able to validate some of these predictions using ENCODE data from shRNA experiments followed by RNA-seq (e.g. <https://www.encodeproject.org/experiments/ENCSR256PLH/>).

This is an excellent suggestion that we have added to our analysis. We compared RNA-seq from control vs shRNA depletion of QKI in HepG2 cells generated by ENCODE and asked how many of the cerebellar exon inclusion or exclusion events showed evidence of a splicing change upon QKI knockdown. Cerebellar regulated exons overall were 5.6 times more likely to change upon QKI knockdown (absolute Δ PSI \geq 20% with $p < 0.05$) versus cerebellar non-changing exons. Moreover, we considered the direction of change upon QKI and found an enrichment of both cerebellar inclusion and cerebellar exclusion events with a shQKI induced change consistent with our model (i.e. QKI depletion promotes the cerebellar splicing pattern, fold enrichment versus non-changing > 4.2 , two-tailed binomial $p < 6.4 \times 10^{-11}$). We did not find a significant enrichment of events inconsistent with our model (i.e. QKI depletion opposes the

cerebellar splicing pattern, fold enrichment versus non-changing <1.3 , two-tailed binomial $p>0.44$). We summarize these results in a new panel to a supplemental figure (Supplementary Fig. 11k in the revised manuscript) and describe this analysis in the Results and methods sections.

Minor concerns:

The low run time of MAJIQ is obviously a nice feature. However, it would be helpful to at least have rough breakdown between the time for MAJIQ Build and Quant to get a better idea of how much time is saved by the incremental build strategy.

That is a good point. Measuring walltime we find the builder (which is run once) is ~ 6.6 times slower than the quantification process. We added information about that in the main text.

It is always hard to summarize and represent the performance of different methods. However, I think it would be a good idea to include either F1 or MCC in Fig 2b to also have a measure which combines more aspects of the confusion matrix than FDR and FNR does.

We agree with the reviewer that having additional stats can be informative. We note though that unlike “traditional” classification tasks where the set of samples/events is fixed, here we have a situation where each method does not report results for the same set of genes (Fig2b, summing the total-change and total-no-change bars). We added the new stats to the bar charts in Fig2b, added Supplementary Table S5-8 which lists all the stats for both gene and event level and both 10% and 20% thresholds (see Reviewer #2 comments below), and added a comment that these stats are not computed over the same set in the caption/main text.

It would be helpful to have labels on the axes in 2d

Done

Some of the text in 3a, b is really small. I had to zoom in on the pdf and this should not be a requirement to be able to read the paper.

We apologize for making the figure unreadable and hope the new version is suitable.

On p17, line 4-5 there are two downstream but one of them should probably be changed to upstream.

We thank the reviewer for noticing this mistake. It has been corrected in the revised manuscript.

It is not for me to decide on these matters, but in my opinion I find the discussion to be on the long side and it is my recommendation to shorten it.

We thank the reviewer for this comment and have worked to shorten/tighten the discussion in this new version. Specifically, the discussion now is over 40% shorter, roughly 820 words total.

On p29 it is not clear how the three priors for dPSI are weighted. Are they all given the same importance?

The prior for dPSI is fit with a mixture of beta distributions. We initialize the mixture components such that the estimated variance of each beta distribution component is computed using only a subset of samples for each: $|\text{dpsi}| < 7.5\%$ for the spike, $7.5\% < |\text{dpsi}| < 32.5\%$ for samples with small/moderate dPSI changes and the rest for the “slab” component. In MAJIQ V1 we used to run EM for a few iterations to update the mixture components but we found this was not improving results compared to just initializing each prior component equally ($\frac{1}{3}, \frac{1}{3}, \frac{1}{3}$). We clarified these details in the supplementary material. Finally, we note that a future update will improve on the above procedure with EM like iterations method of moments to fit beta distribution parameters. This is not fully tested yet as part of another update so is not in the version/description presented here.

Reviewer #2 (Remarks to the Author):

Review of NCOMMS-22-07149

In the manuscript, entitled "RNA splicing analysis using heterogeneous and large RNA-seq datasets", Vaquero-Garcia et al. present an update to their previously published splicing software, MAJIQ, showcasing new tools applied to analyze splicing across large cohorts of RNA-seq samples. These, among other things, are designed to elegantly handle the problems associated with thousands of unannotated minor splicing events, which is a challenge in the field. They compare their software to existing methods, and show that there is significant benefit to using their method which is tailored specifically to the task of comparing across many diverse samples. While the specific statistical tests they benchmark are previously described, and the MAJIQ v2 software is built upon their prior publication, I think the suite of tools they present here has sufficiently diverged from the original publication enough to warrant interest from a broad readership as to its new capabilities and applications.

Overall, I also feel that the analytical and benchmarking contributions presented here are valuable and will be of interest and an asset to the field.

We thank the reviewer for this favorable view. We have invested a monumental amount of work, by far the largest effort to have come out of our lab, in developing and assessing MAJIQ V2. It is gratifying to see that reviewers can appreciate the substantial divergence from the original work as well as the usefulness for the community.

That said, there are several analyses related to Fig. 2 that could be strengthened and/or require changes in order to support the author's claims (e.g. see #3, #7 and #10 below), and additional points for discussion that should be clarified and included or addressed prior to publication (e.g. see #8 & #9).

Please find my list of specific comments below:

(1) I applaud the authors for formalizing their benchmarking analysis as a package for the community. I think it speaks to their efforts to make fair comparisons, enable reproducible science, and move the field forwards in a positive direction. However, at the time of this review the link to `bitbucket.org/biociphers/validations-tools` was not public. It would be helpful if this code could be made available to the reviewers, so that any technical details within it absent from the methods text can be checked.

We deeply apologize for this embarrassing issue. It was never our intention to leave the package unavailable and we actually pointed several researchers to use it in the past so it's unclear when/how this happened. In any case we made sure the package is open/available as it should have been originally and apologize again for this.

(2) For example, in Fig. 2 some of the evaluated programs quantify tandem alternative poly-adenylation sites or tandem alternative promoters, while others do not. (A) Are these event types being simulated in the synthetic data experiments? (B) If not, are these events filtered prior to evaluation to ensure a more stable and fair comparison? The authors should include these details in the methods.

This is an important point and we agree we did not do a good enough job elaborating on this. Regarding (A): The synthetic data was constructed based on whole (known) transcripts quantification of each sample by RSEM. Specific types of events were not added/removed but naturally occurred from that simulation scheme. Regarding (B): We now detail in a separate supplementary subsection how we ran each method in the various tests we applied and if we were not sure how to run a method what tests we performed. This is also discussed in more details below. Specifically for the question raised here: Our understanding is that Whippet defines these transcription end (TE) and transcription start (TS) somewhat differently than other programs (not dependent only on splice junction), so it may capture unique types of events that other methods don't consider. It also searches for back-splicing (BS) which other programs do not. Similarly, MAJIQ is the only method that reports de-novo IR. Thus, to try and achieve "a more stable and fair comparison" we converged to a procedure where when gene levels stats are reported/compared the above unique event types are discarded to create more of "apples to apples" comparison (e.g. Fig2b,c), but when event based analysis is done (Fig 2d,e) these are retained. We did however filter against BS as (a) these seem to be very different and in any case (b) we found these were extremely rare (didn't actually observe them in files we looked at). Inline with the reviewers suggestion we also revisited how we filtered Whippet's events. We originally only included simple binary events ($K < 3$) as Sterne-Weiler et al did in their evaluations. Sterne-Weiler et al were aiming to again create an "apple to apple" comparison with other methods incapable of capturing complex events. In contrast, in this work most methods are capable of capturing complex events and it is part of their contribution. As you will see below, including complex events ($K > 1$) also seemed to give better stats for Whippet. We included descriptions of tests we did for possible run configurations for the various programs in Supplementary Methods, Supplementary Fig 2,3,4,6,7 and additional figures included below.

(3) In the author's comparison of runtime/memory between software packages (Fig. 2a), they compare programs that start from raw-sequencing reads in `fastq` format alongside those that start from pre-aligned reads in `bam` format, without including the alignment time for producing the bam-file in this comparison. For example, Whippet's algorithm is based on internal alignment of junction flanking k-mers from raw-reads directly without external alignment software, while MAJIQ, LeafCutter, and rMATs all require pre-alignment-- a step the authors perform with STAR. Since both the runtime and memory requirements of this alignment step are frequently the most significant computational resources required for the entire start-to-finish RNAseq analysis process-- this discrepancy is a very relevant one. For example, in the text Whippet and SUPPA are presented as the two slowest algorithms (at 50-55 hours vs. 6 hours for other programs), whereas when starting from the same point of the RNAseq analysis in a fair head-to-head evaluation, Whippet and SUPPA are actually much faster than the methods requiring pre-alignment with STAR (assuming SUPPA is leveraging Salmon or Kallisto's speed; see 10.1016/j.molcel.2018.08.018).

Furthermore, it is standard in the field to include this alignment time in such comparisons, as can be seen in previous literature. For example, in the landmark papers for Sailfish: 10.1038/nbt.2862, and Kallisto: 10.1038/nbt.3519, external alignment-time is included in the analogous benchmarks of gene expression quantification software starting from fastq-files (Sailfish/Kallisto/Salmon) when compared to programs that only perform the last RNAseq quantification step starting from pre-aligned bam-files (RSEM, Cufflinks, eXpress etc.). This is also how 4 of the 5 splicing programs evaluated in the current manuscript (eg. MAJIQ, Whippet, SUPPA, rMATs) have been compared in the literature previously (see above).

For the sake of transparency, consistency, and correctness, the authors should: (A) make clear the discrepancy in runtime/memory required for fastq-file vs. bam-file starting-points among programs in both the text and the figures, and (B) include the missing alignment runtime/memory of a standard aligner like STAR for a fastq-file starting point in the final comparison.

This is a fair criticism, and we very much agree we didn't reflect these important differences in the original submission. That said, we would also like to, respectfully, partially push back on the conceived need to have alignment time as an integral part of the methods. First, we note that alignment is used as the common first step for other analysis tasks (e.g. detecting novel genes/exons, APA, gene expression analysis). Regardless, aligned reads are nowadays available as standard output files to download, especially in the context of large datasets (e.g. GTEX) as discussed here. In contrast, the papers referenced by the reviewer are about tools that are designed to perform (pseudo) read alignment, similar to Whippet. That said, the reviewer is absolutely right that the different methods are indeed performing different tasks. This is true btw not only for Whippet but for MAJIQ as well, as it is the only one that detects and quantifies intron retention, a computationally intensive task. Thus, to address the reviewer's

concern we added in the main text a clear discussion of these differences, including estimated times for STAR alignment to take into account.

(4) In Table S4, the authors appear to utilize multicore processing with `number_of_cores=16` for [MAJIQ, rMATs, and LeafCutter], while [SUPPA and Whippet] do not support this. The authors should clarify whether the runtime benchmarks in Fig. 2a are "wall" time (ie. straight start-to-finish clock time ignoring the difference in # cores used) or "cpu" time (ie. additive cpu-time to account for differences using multiple cores). Given the scope of the manuscript towards analyzing large cohorts of heterogeneous samples (which could be run concurrently across multiple cores using single-core programs), multi-core capable programs should not be given any advantage in this manuscript, and "cpu" time should be used.

We respectfully disagree with this assertion. It's true one could write scripts to distribute jobs, and capable users likely will. But we do not think the paper task is to assess or assume such scripts that one could write to distribute jobs across multiple cores in specific contexts. Furthermore, even in a standard laptop nowadays 4 to 8 cores are common. We also note that due to differences in implementation different programs scale differently as a function of the number of cores used. Thus, in our view, running with a "representative" number of cores is a more realistic test for what users can expect to see when running the software "as is". With that said, we agree with the reviewer that those differences between programs were not reflected in the original submission. We thus added text/captions to highlight these differences. Furthermore, we also added tests with only 4 cores to capture a more "representative" usage on a modern laptop compared to the 16 cores desktop.

(5) It is unclear what the author's intention is with the Pg. 7 statement, "0.5-4Gb of memory, an amount readily available on a laptop", or how it fits into the manuscript. Following comment #3 above, the main computational step of start-to-finish RNAseq analysis that might not be compatible with a laptop, would be the alignment step, which the authors excluded (eg. STAR has considerable runtime and requires >~30Gb of RAM, which most laptops cannot accommodate). Furthermore, the benchmarks in the manuscript were performed on a compute node with 32 cores (Intel Xeon 2.7GHz and 64GB RAM) rather than a laptop, obfuscating the statement.

Again, this is a fair point, also a continuation of the previous points raised. Inline with our previous responses above we added information about STAR alignment and memory usage in the revision and highlighted the difference in tasks performed by the methods.

(6) In Fig. 2b, the results look to be very consistent for each program across the six sets of groups: 5v5 through 50v50. If these are essentially redundant data, it might be worth plotting only a subset of these (like 5v5 and 50v50)?

We agree that for brevity of the main manuscript some of the six should be removed from Fig2b. We now only include 3 sets (3vs3, 15vs15, 50vs50).

(7) In Fig. 2b, I think FDR & FNR used here are important metrics to evaluate sensitivity and specificity (inversely). However, one expectation is that differences in evaluated performance might exist as a function of the specific threshold of significance used for each program. Have the authors considered plotting these metrics as a function of a sliding cutoff from a rank-based metric, such as deltaPSI? For example, this additional analysis would importantly reveal the effect of altering significance thresholds towards the outcome, and might more closely resemble standard practices from other areas of classification method evaluation (eg. ROC curves)? This would strengthen the author's claims that MAJIQ's methodology, rather than a specific threshold choice, is responsible for its elevated performance.

The reviewer raises a good point here about sensitivity of the results to the thresholds used. We note that there is a significant difference between the task at hand and that of standard classification with matching ROC curves. The latter is based on fixed labels and a varying score threshold by the software, producing the aforementioned curve. Here, the threshold needs to be applied to both the label and the "score". Furthermore, in standard ROC the set of samples is fixed while here different methods actually report on different sets (those included in the total-change and total-no-change bars in Fig. 2b). Arguably, some of the utility of ROC curve to assess accuracy as a function of a practical threshold applied to the results is fulfilled in our setting by the RR curves, which is a function of #events reported as significant. Finally, the thresholds used here are not arbitrary and reflect biology and common practices regarding what constitutes a "significant change". Nonetheless, we agree it is important to show results are robust to other sensible thresholds. We therefore repeated the analysis for dPSI 10%, added the F1 and MCC metric, reporting these results in the updated Fig2b and and new Tables S5-8.

(8) For Fig. 2b-e & also in the discussion, the authors acknowledge the effect of significance threshold choice in the analysis of LeafCutter, claiming deltaPSI >20% improves its performance in their evaluations of RR. However, it is not fully clear (A) whether they applied non-default parameter adjustments/filters to any of the other algorithms (like rMATS, SUPPA or Whippet), and if so, (B) whether or not these changes affected the performance compared to default? (C) Similarly, if all programs had the same deltaPSI >20% and probability/statistical thresholds-- were these always consistent with the author's documented guidelines/suggestions, or did they diverge for some programs? For example, a deltaPSI cutoff of >20% may improve LeafCutter and MAJIQ, but (D) could it decrease the performance of another program whose statistical test is already inherently more stringent?

While I do not believe it is the benchmarking author's responsibility to try an extensive set of non-default parameters for the competing programs they are evaluating, it would seem to be an important discussion/analysis point for the text-- particularly if some programs (like LeafCutter) are investigated both within and outside of defaults, but others are not (rMATs, SUPPA, and Whippet), or if some programs benefit in performance from non-default thresholds used, but others regress?

The reviewer raises an important point, one that goes well beyond the specific details of our evaluations or even this paper. It has to do with fair/accurate representation of software in

comparative evaluations and where the line should be drawn for authors conducting such evaluation efforts in terms of trying/optimizing other software. We are glad to see the reviewer share our view that it's not the "author's responsibility to try an extensive set of non-default parameters for the competing programs they are evaluating" yet at the same time pointing out the need for a fair/representative analysis.

In general, we take the following approach: First, we strive to use up to date software. Second, if we run into specific problems/issues we are unclear about we contact the authors directly. During our work we ended up contacting the authors of Whippet, LeafCutter, and SUPPA and updated our evaluations several times as new versions/bug fixes were released. We generally gravitate to using software according to the authors specifications unless this seems problematic for the specific assessment task. If that occurs, we try to assess sensible alternatives and report the best performance we find. Our efforts with respect to each methods configuration are reported in a new Supplementary section, Supplementary Fig. 6,7, Tables S5-8, and additional figures included in the description below:

LeafCutter's default execution seems to have been motivated by sQTL analysis and did not include any explicit threshold on dPSI, only on the calculated p-value. We noticed very high false positives (reflected through high IIR values) when using this approach. Furthermore, for tasks where the ground truth is set by defining a threshold on dPSI it seems inappropriate to use a p-value only threshold by LeafCutter when its output also includes dPSI. We therefore deviated from the original authors' execution and report results for events that pass both p-value and the dPSI threshold. When it comes to ordering events (as required for RR plots) we again found ourselves without author specified guidelines. We therefore tried both options that pass both filters: order by dPSI or order by p-value. We report results for the setting which resulted in better performance.

For rMATS, we similarly applied both filters (p-value and dPSI), tried ranking by either, reporting the better results.

For SUPPA, we also applied both filters (p-value and dPSI) and tried ranking by either, reporting the better results.

For all of the above we used the standard p-value cutoff ($p\text{-value} < 0.05$) and we tested two different dPSI thresholds commonly used in the RNA splicing field: 10% and 20%.

For Whippet, we ended up needing to test several different configurations. First, after contacting the authors we found that the evaluations performed in Sterne-Weiler et al applied the following filters: First, events were required to have a CI < 0.1 in ALL samples. The CI (confidence interval), serves as a measure of how sure Whippet is in the quantification. For small datasets, typically composed of biological replicates, these thresholds make sense as these should leave users with a confidently quantified set. However, for large heterogeneous data as assessed here, we found these thresholds to be overly restrictive as many AS events may end up not passing the CI < 0.1 in at least some of the (many) samples. Left as is, these thresholds result

in poor performance. We therefore left Whippet confidence parameter per event in a sample as is ($CI < 0.1$) but required only 50% of the samples to pass that filter - analogous in spirit to MAJIQ's default filter of requiring an event to be quantifiable in at least 50% of the samples in the group. We also tried an 80% threshold but 50% seemed better.

Another consideration in Whippet's testing regards the complexity of the events reported by Whippet. Whippet uses the parameter K to denote the number of possible paths through the AS event. A classical event (e.g. cassette exon) would have $K=2$, while more complicated events would have $K > 2$. In Sterne-Weiler et al the authors evaluated Whippet (and all other competing methods) only on $K < 3$ events, i.e. only on simple "binary" events. This decision stemmed from the desire to create a common base with methods such as rMATS which only operate over classical (binary) AS events. In the original submission of this manuscript we followed the same restriction of $K < 3$ as in Sterne-Weiler et al. However, given the reviewer's point we re-tested Whippet with both complex and binary (i.e. no $K < 3$ filter). Our rationale for this was that many methods (Whippet, LeafCutter, MAJIQ) can now handle complex events. Furthermore, it is important to include those since in many cases the ability to detect and report such events is part of the highlights/selling points of such methods and users should be able to assess performance in the context of what the method actually gives them (i.e. both complex and binary events). As can be expected, Whippet's metrics with no filtering of $K > 2$ improved in terms of the total number of genes reported in Fig. 2b, as shown here (compare "Whippet" to "Whippet Complex" which removes the $K < 3$ filter, note this is a summary at the gene level) :

At the same time, we did not observe a significant difference when assessing overall dPSI accuracy at the event level (compare Whippet on the left with “Whippet Complex” on the right for events reported with dPSI > 20%, notice the substantial increase in the number of events reported). To avoid misunderstanding, we note the plots below are done at the event level and not the gene level as the one above, hence the numbers are much higher.

The above plots for whippet without complex events were not included in the supplementary figures as these are already packed, but we include those here for completeness (to clarify: The supplementary figures do include of course the CDF plots for the methods as above, but only showing “Whippet Complex” which is simply termed “Whippet” there).

Yet another consideration when evaluating methods that detect very different event types is which types of AS events to include/exclude. For example, Whippet detects events of type transcriptional start (TS), transcriptional end (TE), and back-splicing (BS), which other methods do not. Similarly, MAJIQ is the only method that detects novel intron retention. After contemplating much about how such differences should be considered we converged to the following procedure: In tests where we use the methods “native language” of AS events it reports we kept all event types (but see restriction applied against MAJIQ to avoid “double counting” and inflating its stats, as described below). This applies to RR and IIR where the logic is to assess each method by its own set of events in terms of what the user will get. However, when trying to create a more uniform assessment at the gene level rather than at the event level, as we did using synthetic data, then we excluded unique event types (i.e. Whippet’s TS/TE/BS and IR). In all cases we did not exclude complex events as most methods detect those (see also discussion above) and the criteria applied is a change by 10% or 20% based on the methods own event “language”.

For MAJIQ we didn’t perform any special tweaks/parameter optimization. We ran it with default parameters, with the only exception being that we used the flag “show-all” in order to actually get in the output files ALL events assessed by MAJIQ rather than only the events that pass the default (conservative) thresholds of dPSI > 20% with a posterior probability > 0.95. We note that failing to using this flag while trying to assess MAJIQ’s performance over the entire set of events at different thresholds (i.e. using default output but not as intended for regular use) has led to severe misrepresentation of MAJIQ’s performance as we have documented in detail in an earlier pre-print (Vaquero et al 2018).

The only tweak we performed for MAJIQ's evaluation is to exclude overlapping LSVs when assessing performance at the event level (i.e. for RR and IIR). This is because classical events reported by other software (e.g. rMATS) can be captured by overlapping LSVs such that counting/reporting those would inflate MAJIQ's stats unfairly. For example, a highly significant change in a cassette exon detected by rMATS would be captured by two LSVs in MAJIQ: a single source LSV and a single target LSV. Thus, when reporting RR and IIR we rank all LSV as usual but an LSV is disregarded if an overlapping LSV has already been included in the RR curve. The effect of not filtering overlapping LSV (increase ~50% number of LSV reported while maintaining very similar RR) is reported in Supplementary Fig. 7.

To summarize, we didn't try to exhaustively test all parameter combinations for any specific software (MAJIQ included). When using default parameters did not make sense or did not match the test's assumptions we tried to give each method a fair representation, exploring additional parameters/filter settings to the best of our understanding, and reporting the best results. Finally, we note that all our testing data, test scripts and parameters are documented and even included as a package for future use.

As a post-script to this comment -- I'd like to point out that Whippet-documentation for ``delta.jl`` provides no more than a loose guideline on significance criteria (probability > 0.9 and absolute deltaPSI > 0.1), with a note that while the quantification of PSI with ``quant.jl`` has been extensively benchmarked for accuracy, for differential splicing "no specific significance thresholds have been systematically optimized". Given the evaluations presented in Fig. 2b-e, it would seem that this lack of calibration negatively affects the out-of-the-box performance of ``delta.jl``, and I commend the authors of Vaquero-Garcia et al. for their analyses, which point out this limitation.

We thank the reviewer for pointing out our efforts in this regard.

(9) In the manuscript, previously described statistical methods are applied to MAJIQ in order to identify significant differences in MAJIQ's underlying PSI values across large heterogeneous sample sets (TNOM, InfoScore and Mann-Whitney U). Since some of these statistical methods are currently available with external implementations (eg. Mann-Whitney-U is a base function in most languages, and TNOM looks to be implemented in the ClassComparison R package), they could also be easily applied to any of the other methods compared in this manuscript (eg. LeafCutter, SUPPA, rMATS, Whippet). For example, in Fig. 4 the authors define significant events using a simple Wilcoxon test (which is standardly applied across research in many fields, including splicing).

While that choice of statistical test may not be unique to MAJIQ here, it is likely that the performance of any statistical test in question will be limited by the accuracy of the underlying single-sample splicing quantifications used (eg. PSI values). This makes the accuracy of single-sample splicing quantifications an important metric in evaluating the advances MAJIQ provides in relation to other software methods of comparable scope. The authors should consider including this as a discussion point.

This is a good point raised by the reviewer. The reviewer is right that one could potentially apply different statistical tests atop whatever quantification method of PSI or dPSI is used. While assessing all potential combinations/tests is well beyond the scope of this paper it is an interesting question to ponder: Are the differences between the methods simply the result of a different statistical test? To try and at least partially address that question we include two types of tests. First, we include accuracy assessment using a set of measurements from an independent experimental technique - triplicates of RT-PCR experiments (also see below). However, while valuable as an independent measure these RT-PCR experiments only cover a limited number of AS events which are also biased. Specifically, this set of validated events were selected to include high changes between tissues and only binary non-denovo (i.e. annotated) events so that methods such as rMATS could detect those as well. Thus, we looked for a second type of measure to include in this revision to address the reviewer's point. Since the question was about the effect of the specific statistical test on the result we decide to simply take those out: Using the GTEx simulated data, we plotted for each method a CDF over all the events it reports as significantly changing (here too, we repeated this test for two different thresholds - 10% and 20%). The CDF for each method was computed over the absolute deviation between the estimated dPSI and the "true" dPSI of the group, for different group sizes. We note that since no statistical test is applied this test addresses the overall accuracy of the method directly, as asked by the reviewer. The results of this analysis are reported in Supplementary Fig. 4.

(10) In Supplemental Fig. S2, (and the text on Pg. 10), the authors do compare the accuracy of single-sample PSI point-estimates to experimental values from RT-PCR for MAJIQ and LeafCutter, but do not include plots of these accuracy values side-by-side to the other programs benchmarked in Fig. 2 (eg. SUPPA, rMATS, Whippet). While the authors refer to their previous publication where they plot these values for SUPPA and rMATS, Whippet is not included. (A) The authors could plot a summary of this data and discuss whether or not accuracy at this level (or lack thereof) is concordant with their results when benchmarking deltaPSI in Fig. 2.

Similarly, given the importance of single-sample splicing quantification accuracy (see #9 above), (B) the authors should assess the accuracy of single-sample PSI using the synthetic data they simulated for Fig 2 -- plotting PSI point-estimate vs. ground-truth PSI (summary panels of these analyses could perhaps join the others in Fig 2).

We now include RT-PCR assessment for all methods as well as assessment whether differences are statistically significant (see Reviewer #1 comment above, Fig. 2f, Supplementary Fig. 5). We now include assessment of PSI accuracy as it relates to dPSI assessment (the main aim of these methods) as described in the previous section, Supplementary Fig. 4.

(11) Fig. 3 -- I think the Voila Modulizer and visualization tools look great. In general, visualization of splicing for large heterogenous cohorts is a weak point in the field, and a tool like this has a lot of value and should accelerate research. Towards this end-- the discussion mentions that a lab or consortia center could process data like GTEx once, and then add their

own samples to it from future experiments. Since the one-time analysis of the entire GTEx cohort is a considerable endeavor in and of itself, another point to highlight here might be how easy it is for a single user to compare a few internally produced samples of interest side-by-side to all of the pre-analyzed GTEx data (either by uploading their data to a remote server that contains the data, or by downloading the pre-analyzed data if it is made available)?

We thank the reviewer for raising here additional suggestions to make our work more useful! Since the GTEx consortium made junction read files per sample available now on its website we can now safely release MAJIQ files for all the brain subregions samples analyzed in this study as a resource (see updated data and code availability section). Users will be able to upload our pre-processed MAJIQ files and combine them with their own data or other datasets of interest. We also like the idea of enabling users to upload and compare samples against GTEx onto a server but that requires establishing one with search/upload capabilities etc. Such a tool is well beyond the scope of this work but we would be looking into creating it as a resource.

References

Braunschweig, U. et al. Widespread intron retention in mammals functionally tunes transcriptomes. *Genome Research* 24, 1774–1786 (2014).

Jiang, L., Wang, M., Lin, S., Jian, R., Li, X., Chan, J., ... & Doherty, J. A. (2020). A quantitative proteome map of the human body. *Cell*, 183(1), 269-283.

REVIEWER COMMENTS

Reviewer #1 (Remarks to the Author):

I would like to congratulate the author who was on parental leave, hoping that they are able to get some restful nights of sleep.

Overall, I believe that the authors have done a very good job, and all of my comments have been addressed. I did however, note some minor issues:

P17: it says E(PSI) in the text, but I think you should use greek letter, square brackets and fat E?

P17 (bottom) "were stability detected" -> "were stably detected"

Reviewer #2 (Remarks to the Author):

The authors have fixed many of the issues I raised in the initial submission and I expect their manuscript will represent a valuable contribution to the field once published. However, a few outstanding areas still need to be addressed to satisfy the points of contention in the original review. Specifically, at present, I still feel that the figure (and text) with respect to speed benchmarks (shown in Fig. 2a) may be misleading to readers who expect all programs to have the same starting points. Here I qualify some of my original points with additional explanation/examples, and ultimately provide specific suggestions of changes that I think the authors could make. Reasonable additions, such as including both the 'fastq' file starting point and 'cpu time' measurement alongside the 'bam' file starting point and 'wall time' rather than substituting for them entirely, would both increase the fairness and transparency of comparisons to existing programs while still ensuring the authors can highlight the specific context in which MAJIQ performs favorably to existing tools.

``` Original Reviewer Comment #3

(3) In the author's comparison of runtime/memory between software packages (Fig. 2a), they compare programs that start from raw-sequencing reads in `fastq` format alongside those that start from pre-aligned reads in `bam` format, without including the alignment time for producing the bam-file in this comparison. For example, Whippet's algorithm is based on internal alignment of junction flanking k-mers from raw-reads directly without external alignment software, while MAJIQ, LeafCutter, and rMATs all require pre-alignment-- a step the authors perform with STAR. Since both the runtime and memory requirements of this alignment step are frequently the most significant computational resources required for the entire start-to-finish RNAseq analysis process-- this discrepancy is a very relevant one. For example, in the text Whippet and SUPPA are presented as the two slowest algorithms (at 50-55 hours vs. 6 hours for other programs), whereas when starting from the same point of the RNAseq analysis in a fair head-to-head evaluation, Whippet and SUPPA are actually much faster than the

methods requiring pre-alignment with STAR (assuming SUPPA is leveraging Salmon or Kallisto's speed; see 10.1016/j.molcel.2018.08.018).

Furthermore, it is standard in the field to include this alignment time in such comparisons, as can be seen in previous literature. For example, in the landmark papers for Sailfish: 10.1038/nbt.2862, and Kallisto: 10.1038/nbt.3519, external alignment-time is included in the analogous benchmarks of gene expression quantification software starting from fastq-files (Sailfish/Kallisto/Salmon) when compared to programs that only perform the last RNAseq quantification step starting from pre-aligned bam-files (RSEM, Cufflinks, eXpress etc.). This is also how 4 of the 5 splicing programs evaluated in the current manuscript (eg. MAJIQ, Whippet, SUPPA, rMATs) have been compared in the literature previously (see above).

For the sake of transparency, consistency, and correctness, the authors should: (A) make clear the discrepancy in runtime/memory required for fastq-file vs. bam-file starting-points among programs in both the text and the figures, and (B) include the missing alignment runtime/memory of a standard aligner like STAR for a fastq-file starting point in the final comparison.

...

``` Author Response to #3

This is a fair criticism, and we very much agree we didn't reflect these important differences in the original submission. That said, we would also like to, respectfully, partially push back on the conceived need to have alignment time as an integral part of the methods. First, we note that alignment is used as the common first step for other analysis tasks (e.g. detecting novel genes/exons, APA, gene expression analysis). Regardless, aligned reads are nowadays available as standard output files to download, especially in the context of large datasets (e.g. GTEX) as discussed here. In contrast, the papers referenced by the reviewer are about tools that are designed to perform (pseudo) read alignment, similar to Whippet. That said, the reviewer is absolutely right that the different methods are indeed performing different tasks. This is true btw not only for Whippet but for MAJIQ as well, as it is the only one that detects and quantifies intron retention, a computationally intensive task. Thus, to address the reviewer's concern we added in the main text a clear discussion of these differences, including estimated times for STAR alignment to take into account.

...

```Reviewer Response to Authors for #3

Despite text changes made in revision, the lack of alignment time in the benchmarks remains an important outstanding issue. In the spirit of compromise to resolve the discrepancy, one idea might be for the authors to separate the Fig. 2a panels, or represent the same panels with additional colors or shapes— one with a 'fastq' starting point that includes the missing alignment time for the programs that begin from bam files. And then a second could represent the 'bam' file starting point. Such a change would both provide transparency as to the actual comparisons being performed, meanwhile also informing on the performance advantages of starting from pre-aligned 'bam' files using MAJIQ.

Separately, to respond to the author's push back-- I would put forward that the quality of the upstream alignment step is less independent of the accuracy of the downstream splicing analysis by MAJIQ than the authors suggest. For example, Veeneman et al. 2016 (doi: 10.1093/bioinformatics/btv642) found that a two-pass alignment procedure with STAR (as opposed to the default single-pass alignment) improves accuracy over unannotated exon-exon junctions. Such discrepancy in alignment parameters is expected to impact on the accuracy of all downstream quantification of overlapping splicing events (for programs that utilize pre-alignment like MAJIQ, LeafCutter, and rMATs). Following this, of the other two programs starting from STAR alignments evaluated in this manuscript, (a) LeafCutter (<https://davidaknowles.github.io/leafcutter/articles/Usage.html>) actually provides specific instructions for this two-pass alignment as the first step to running their software, and (b) rMATs (<https://github.com/Xinglab/rmats-turbo/tree/v4.1.2>) has two formal input file formats, one of which is 'fastq' where rMATs internally runs the alignment step with STAR using their own specific parameters. Both of these examples imply that the alignment step is an integral part of their methodology, contrary to the author's assessment.

I would also like to point out that the same two-pass alignment with STAR was performed in this paper, meanwhile the standard default single-pass alignment is both faster and still suitable for other more common tasks such as gene expression analysis. Separately, while I do note that the author's point is fair about the existence of pre-aligned reads in 'bam' format for large datasets (eg. GTEx), it appears that the GTEx pipeline performs only the default single-pass alignment with STAR (as far as I can tell from the pipeline's official github: <https://github.com/broadinstitute/gtex-pipeline/tree/master/rnaseq>). Therefore, it remains unclear how much error in resulting PSI a downstream splicing analysis might have that started from this data – a relevant discrepancy that the authors don't investigate here. In fact, in the revised text, the authors use the existence of such single-pass pre-alignment files as reasoning to not include the time required to produce those alignments in their speed benchmarking, meanwhile assessing the accuracy of their method on higher quality alignments.

...

``` Original Reviewer Comment #4

(4) In Table S4, the authors appear to utilize multicore processing with number_of_cores=16 for [MAJIQ, rMATs, and LeafCutter], while [SUPPA and Whippet] do not support this. The authors should clarify whether the runtime benchmarks in Fig. 2a are "wall" time (ie. straight start-to-finish clock time ignoring the difference in # cores used) or "cpu" time (ie. additive cpu-time to account for differences using multiple cores). Given the scope of the manuscript towards analyzing large cohorts of heterogeneous samples (which could be run concurrently across multiple cores using single-core programs), multi-core capable programs should not be given any advantage in this manuscript, and "cpu" time should be used.

...

``` Author Response to #4

We respectfully disagree with this assertion. It's true one could write scripts to distribute jobs, and capable users likely will. But we do not think the paper task is to assess or assume such scripts that one could write to distribute jobs across multiple cores in specific contexts. Furthermore, even in a standard laptop nowadays 4 to 8 cores are common. We also note that due to differences in implementation different programs scale differently as a function of the number of cores used. Thus, in our view, running with a "representative" number of cores is a more realistic test for what users can expect to see when running the software "as is". With that said, we agree with the reviewer that those differences between programs were not reflected in the original submission. We thus added text/captions to highlight these differences. Furthermore, we also added tests with only 4 cores to capture a more "representative" usage on a modern laptop compared to the 16 cores desktop.

...

``Reviewer Response to Authors for #4

I would like to qualify my original comment with additional explanation and examples. Specifically, a multi-threading program (ie. MAJIQ on a computer with 4 cores) can analyze a single sample across the available CPUs, making a task that would take time  $\{t\}$ , finish nearly 4x faster, theoretically finishing in just over  $\sim\{t/4\}$  time. However, if a user wanted to analyze four different samples or replicates, they would either have to run them at the same time (ie. concurrently) or one after another (consecutively). If concurrently, they would each compete with one another for resources, obtaining only  $\sim 25\%$  of each CPU and taking  $\sim 4$  times as long as they would alone—therefore nullifying the speed advantage of using multiple cores,  $\{t/4\} * 4 = \{t\}$ . If consecutive, the  $\{t/4\}$  runtimes add together to yield the same result, also  $\{t/4\} * 4 = \{t\}$ . Similarly, if a single-core program (ie. Whippet or SUPPA on a computer with 4 cores) with the same runtime  $\{t\}$  was run concurrently on those same four samples and machine with 4 cores, each one would use 100% of CPU (one sample per cpu-core), and all four would also finish in  $\{t\}$  time.

Since most (if not all) scientific investigations require multiple samples or replicates (which is also the focus of this manuscript), the discrepancy of how programs utilize available compute resources wouldn't ultimately have much effect on multi-sample analysis performance (whether single samples consecutively multithreading across multiple cores, or multiple samples run concurrently on single-cores). Yet, by using "wall time" with a different number of cpu-cores assigned to each program, the author's analysis provides arbitrarily large speed gains (originally 16-fold, now they have lowered it to 4-fold) to programs with multithreading capability as compared to programs without. Such a comparison where the authors can "choose" the multifold amount of speed up some programs get compared to others based on what they consider "representative" seems overly subjective.

In their response to my review point, the authors incorrectly claim that only a user with advanced capability could make use of such job distribution for single-core programs. However, the same capability required to analyze a single sample, would be suitable to run a reasonable number of additional samples concurrently on a personal computer (just by changing the filename in the command and pressing enter). The downstream assignment of each sample to the unused available cores happens natively in an operating system and requires no additional user knowledge or skills (ie. "it just works").

To compound this statement, for very large sample cohorts (which again, is the focus of this manuscript), it is hard to imagine how distributed cluster or cloud computing is not required, and such systems are based on formal schedulers that dispatch jobs (eg. slurm or qsub). While these systems do require some user knowledge to operate —it is often the same knowledge needed to run a single-job as for hundreds.

Lastly, I want to point out that beyond my original assertion of correctness with using “cpu time” (as opposed to “wall time”) to account for differences in multithreading core usage-- The requested change follows the commonly accepted best practices in the field. In a recent literature review published in Nature Communications by Mangul et al. in 2019 entitled “Systematic benchmarking of omics computational tools”, it is also suggested that “cpu time should generally be used” (doi: 10.1038/s41467-019-09406-4).

In an effort to promote fairness, transparency and rigor, I would like to again, respectfully re-assert that cpu-time should be used instead of wall-time to account for the discrepancy in compute cores. At the very least, the authors should preserve transparency by indicating both cpu-time in addition to wall-time in the figure (and in the text) either as separate colored points/bars/lines, or as adjacent panels.

...

``` New Minor Comment #1:

Methods Pg. 42 “A classical event (e.g. cassette exon) would have $K = 2$, while more complicated events would have $K > 2$.”

The previous statement in methods section for Whippet is a typo/error. $K=1$ is the value for simple cassette exons. Where $K = \log_2(\text{max isoforms})$ for a given splicing event, so for a binary splicing event, $K = \log_2(2) = 1$. Complex events would be $K \geq 2$.

...

We thank both reviewers for taking the time to read through our revised manuscript and send their feedback. We were very pleased to see Reviewer #1 was happy with the updates and we thank them for catching the mistakes they reported - we fixed those in this update.

We would also like to thank Reviewer #2 for their detailed feedback. It was gratifying to see there are no actual scientific issues with our work. To summarize succinctly, the only remaining issue raised by Reviewer #2 is how time/memory performance are presented in Fig 2a for a fair and accurate comparison. Detailed points raised and our response those are listed below.

REVIEWER COMMENTS

Reviewer #1 (Remarks to the Author):

I would like to congratulate the author who was on parental leave, hoping that they are able to get some restful nights of sleep.

Overall, I believe that the authors have done a very good job, and all of my comments have been addressed. I did however, note some minor issues:

P17: it says E(PSI) in the text, but I think you should use greek letter, square brackets and fat E?

P17 (bottom) “were stability detected” -> “were stably detected”

Both of the above errors were corrected and we thank Reviewer #1 again for their time and effort leading to the improvement of our manuscript.

Reviewer #2 (Remarks to the Author):

The authors have fixed many of the issues I raised in the initial submission and I expect their manuscript will represent a valuable contribution to the field once published. However, a few outstanding areas still need to be addressed to satisfy the points of contention in the original review. Specifically, at present, I still feel that the figure (and text) with respect to speed benchmarks (shown in Fig. 2a) may be misleading to readers who expect all programs to have the same starting points. Here I qualify some of my original points with additional explanation/examples, and ultimately provide specific suggestions of changes that I think the authors could make. Reasonable additions, such as including both the ‘fastq’ file starting point and ‘cpu time’ measurement alongside the ‘bam’ file starting point and ‘wall time’ rather than substituting for them entirely, would both increase the fairness and transparency of comparisons to existing programs while still ensuring the authors can highlight the specific context in which MAJIQ performs favorably to existing tools.

For the detailed response below, and given how dense Reviewer #2 response was we tried to group the various paragraphs into subjects/themes without altering the actual content. We also tried to highlight the keypoints using underline, so that the response can be easily connected to those.

Despite text changes made in revision, the lack of alignment time in the benchmarks remains an important outstanding issue. In the spirit of compromise to resolve the discrepancy, one idea might be for the authors to separate the Fig. 2a panels, or represent the same panels with additional colors or shapes— one with a ‘fastq’ starting point that includes the missing alignment time for the programs that begin from bam files. And then a second could represent the ‘bam’ file starting point. Such a change would both provide transparency as to the actual comparisons being performed, meanwhile also informing on the performance advantages of starting from pre-aligned ‘bam’ files using MAJIQ.

I would like to qualify my original comment with additional explanation and examples. Specifically, a multi-threading program (ie. MAJIQ on a computer with 4 cores) can analyze a single sample across the available CPUs, making a task that would take time $\{t\}$, finish nearly 4x faster, theoretically finishing in just over $\sim\{t/4\}$ time. However, if a user wanted to analyze four different samples or replicates, they would either have to run them at the same time (ie. concurrently) or one after another (consecutively). If concurrently, they would each compete with one another for resources, obtaining only $\sim 25\%$ of each CPU and taking ~ 4 times as long as they would alone—therefore nullifying the speed advantage of using multiple cores, $\{t/4\} * 4 = \{t\}$. If consecutive, the $\{t/4\}$ runtimes add together to yield the same result, also $\{t/4\} * 4 = \{t\}$. Similarly, if a single-core program (ie. Whippet or SUPPA on a computer with 4 cores) with the same runtime $\{t\}$ was run concurrently on those same four samples and machine with 4 cores, each one would use 100% of CPU (one sample per cpu-core), and all four would also finish in $\{t\}$ time.

Since most (if not all) scientific investigations require multiple samples or replicates (which is also the focus of this manuscript), the discrepancy of how programs utilize available compute resources wouldn't ultimately have much effect on multi-sample analysis performance (whether single samples consecutively multithreading across multiple cores, or multiple samples run concurrently on single-cores). Yet, by using “wall time” with a different number of cpu-cores assigned to each program, the author’s analysis provides arbitrarily large speed gains (originally 16-fold, now they have lowered it to 4-fold) to programs with multithreading capability as compared to programs without. Such a comparison where the authors can “choose” the multifold amount of speed up some programs get compared to others based on what they consider “representative” seems overly subjective.

In their response to my review point, the authors incorrectly claim that only a user with advanced capability could make use of such job distribution for single-core programs. However, the same capability required to analyze a single sample, would be suitable to run a reasonable

number of additional samples concurrently on a personal computer (just by changing the filename in the command and pressing enter). The downstream assignment of each sample to the unused available cores happens natively in an operating system and requires no additional user knowledge or skills (ie. “it just works”). To compound this statement, for very large sample cohorts (which again, is the focus of this manuscript), it is hard to imagine how distributed cluster or cloud computing is not required, and such systems are based on formal schedulers that dispatch jobs (eg. slurm or qsub). While these systems do require some user knowledge to operate —it is often the same knowledge needed to run a single-job as for hundreds.

Lastly, I want to point out that beyond my original assertion of correctness with using “cpu time” (as opposed to “wall time”) to account for differences in multithreading core usage-- The requested change follows the commonly accepted best practices in the field. In a recent literature review published in Nature Communications by Mangul et al. in 2019 entitled “Systematic benchmarking of omics computational tools”, it is also suggested that “cpu time should generally be used” (doi: 10.1038/s41467-019-09406-4).

In an effort to promote fairness, transparency and rigor, I would like to again, respectfully re-assert that cpu-time should be used instead of wall-time to account for the discrepancy in compute cores. At the very least, the authors should preserve transparency by indicating both cpu-time in addition to wall-time in the figure (and in the text) either as separate colored points/bars/lines, or as adjacent panels.

Trying to summarize the points raised about we found the key issues raised by Reviewer #2 are:

- (a) We should report CPU time and not wall-time as it is a more fair comparison and in any case distributing jobs across CPUs is not an expert task. It’s also more representative of performance and is a common practice suggested by Magul et al 2019.
- (b) Related to that: The Reviewer’s method, Whippet, suffers greatly when reporting wall-time on multiple core machines as it only supports a single thread.
- (c) Related to that: The Reviewer’s method, Whippet, suffers greatly since it is the only method that performs an additional task, read mapping, which other methods assume as a given input.

Regarding (a):

In the revision we kept using wall-time as we believe it is a more realistic representation of what users can expect rather than a theoretic value. We note that even the paper cited by Reviewer #2 (Mangul et al 2019) as an argument states “....*Computational cost of a tool is an important criterion for which there are several means of evaluation....as execution time may vary across different servers, benchmarking studies should report server specifications and number of processors used*”. We do exactly that. Furthermore, CPU time does not take into account other bottlenecks such as I/O that can have a major impact on performance. All and all, we hold there is nothing wrong (and arguably better) with the way we present time/memory. Sending us to create single CPU runs for all methods on large data is a huge waste of time/resources that

serves no real purpose as there is no real dispute about the science but rather on the fairness of the presentation, which is a valid point - leading us to the adaptations we made to resolve the issues raised by Reviewer #2.

Regarding (b)+(c):

These points seem to be the crux of the matter. While we believe we previously included very clear statements about Whippet running on a single core and performing the mapping step, the consequent visuals in Fig2 are clear and disfavor it. We very much understand the Reviewer's pain that a specific method (Whippet) suffers greatly in those plots as a consequence of the assessment approach we took. We accept their point that these conceived gaps can be made arbitrarily large if one method is run on a single core (Whippet) but others do not, making the visuals again greatly disfavoring single core methods even when the disclaimer is included in the text.

Following on Reviewer #2 suggestion we therefore made the following adjustments:

- (a) We implemented a script to distribute jobs across the 16 cores and tested Whippet's performance with this manual parallelization using 4,8,16 cores. 8 and 16 cores performed similarly, likely due to I/O. We report this testing as an additional tune up in the supplementary section with a matching figure. Based on that we ran both Whippet and SUPPA (which we noticed is limited in parallel execution as well) with 16 threads and now report results for those in Fig 2. To make the above change transparent to the reader we labeled the methods as "Whippet(x16)" and "SUPP(x16)" in the figure legend and described this addition in the main text.
- (b) We added time and memory using stacked bars/lines with different colors for both STAR and SALMON (used by SUPPA2).

Separately, to respond to the author's push back-- I would put forward that the quality of the upstream alignment step is less independent of the accuracy of the downstream splicing analysis by MAJIQ than the authors suggest. For example, Veeneman et al. 2016 (doi: 10.1093/bioinformatics/btv642) found that a two-pass alignment procedure with STAR (as opposed to the default single-pass alignment) improves accuracy over unannotated exon-exon junctions. Such discrepancy in alignment parameters is expected to impact on the accuracy of all downstream quantification of overlapping splicing events (for programs that utilize pre-alignment like MAJIQ, LeafCutter, and rMATs). Following this, of the other two programs starting from STAR alignments evaluated in this manuscript, (a) LeafCutter (<https://davidaknowles.github.io/leafcutter/articles/Usage.html>) actually provides specific instructions for this two-pass alignment as the first step to running their software, and (b) rMATs (<https://github.com/Xinglab/rmats-turbo/tree/v4.1.2>) has two formal input file formats, one of which is 'fastq' where rMATs internally runs the alignment step with STAR using their own specific parameters. Both of these examples imply that the alignment step is an integral part of their methodology, contrary to the author's assessment.

I would also like to point out that the same two-pass alignment with STAR was performed in this paper, meanwhile the standard default single-pass alignment is both faster and still suitable for other more common tasks such as gene expression analysis. Separately, while I do note that the author's point is fair about the existence of pre-aligned reads in 'bam' format for large datasets (eg. GTEx), it appears that the GTEx pipeline performs only the default single-pass alignment with STAR (as far as I can tell from the pipeline's official github: <https://github.com/broadinstitute/gtex-pipeline/tree/master/rnaseq>). Therefore, it remains unclear how much error in resulting PSI a downstream splicing analysis might have that started from this data – a relevant discrepancy that the authors don't investigate here. In fact, in the revised text, the authors use the existence of such single-pass pre-alignment files as reasoning to not include the time required to produce those alignments in their speed benchmarking, meanwhile assessing the accuracy of their method on higher quality alignments.

Some of the above comments by Reviewer #2 were made to support including read mapping time/memory which we added now (see previous points/discussion).

However, Reviewer #2 adds here an additional new point about 1 or 2 pass runs for STAR - We strongly object to it. No method evaluated here, except Whippet which has internal read mapping, is handling mapping. STAR 1 or 2 pass mapping issues are completely beyond the scope of this paper. Even running time of an additional second pass can be greatly affected by the parameters used (see for example a discussion here: https://groups.google.com/g/rna-star/c/QxLgZxgOzko/m/EJKk_agACAAJ). We can add that in our testing a second pass can definitely affect detection of novel junctions and some differential events as well, but the overall effect on differential splicing accuracy, the main aim of the tools here, is small. For example, in several ENCODE data (which is expected to be more affected by de-novo junctions than GTEX as these are KD experiments) the 99 percentile of dPSI between 1 and 2 pass was ~2.5% dPSI. While some events may change the overall curves of RR etc. For differential splicing as we evaluate here over GTEX stay stable and do not affect the comparison of the methods. Lastly, in practice, for large studies as assessed here the mapper settings are irrelevant as the vast majority of users will simply download the BAM files already supplied.

``` New Minor Comment #1:

Methods Pg. 42 "A classical event (e.g. cassette exon) would have  $K = 2$ , while more complicated events would have  $K > 2$ ."

The previous statement in methods section for Whippet is a typo/error.  $K=1$  is the value for simple cassette exons. Where  $K = \log_2(\text{max isoforms})$  for a given splicing event, so for a binary splicing event,  $K = \log_2(2) = 1$ . Complex events would be  $K \geq 2$ .

We thank Reviewer #2 for noticing these errors which have now been fixed.

In conclusion we believe the above changes should fully address Reviewer #2 concerns and we hope to be able to move forward with the manuscript.